# Neural variability determines coding strategies for natural self-motion in macaque monkeys

Isabelle Mackrous[1], Jérome Carriot[1], Kathleen E Cullen[2,3,4,5], Maurice J Chacron[1]*

[1]Department of Physiology, McGill University, Montreal, Canada; [2]The Department of Otolaryngology- Head and Neck Surgery, Johns Hopkins University School of Medicine, Baltimore, United States; [3]The Department of Biomedical Engineering, Johns Hopkins University School of Medicine, Baltimore, United States; [4]The Department of Neuroscience, Johns Hopkins University School of Medicine, Baltimore, United States; [5]Kavli Neuroscience Discovery Institute, Johns Hopkins University, Baltimore, United States

**Abstract** We have previously reported that central neurons mediating vestibulo-spinal reflexes and self-motion perception optimally encode natural self-motion (Mitchell et al., 2018). Importantly however, the vestibular nuclei also comprise other neuronal classes that mediate essential functions such as the vestibulo-ocular reflex (VOR) and its adaptation. Here we show that heterogeneities in resting discharge variability mediate a trade-off between faithful encoding and optimal coding via temporal whitening. Specifically, neurons displaying lower variability did not whiten naturalistic self-motion but instead faithfully represented the stimulus' detailed time course, while neurons displaying higher variability displayed temporal whitening. Using a well-established model of VOR pathways, we demonstrate that faithful stimulus encoding is necessary to generate the compensatory eye movements found experimentally during naturalistic self-motion. Our findings suggest a novel functional role for variability toward establishing different coding strategies: (1) faithful stimulus encoding for generating the VOR; (2) optimized coding via temporal whitening for other vestibular functions.

*For correspondence:
maurice.chacron@mcgill.ca

Competing interests: The authors declare that no competing interests exist.

## Introduction

Our previous study was the first to investigate how neurons within the first central stage of vestibular processing in macaque monkeys respond to natural self-motion (*Mitchell et al., 2018*). Specifically, we focused on a class of neurons within the vestibular nuclei, vestibular-only (VO) neurons, that mediate vestibulo-spinal reflexes as well as self-motion perception (*Abzug et al., 1974*; *Shinoda et al., 1988*; *Gdowski and McCrea, 1999*; *Meng et al., 2007*; *Marlinski and McCrea, 2009*). Our results revealed that VO neurons optimally encoded naturalistic self-motion stimuli through temporal whitening (i.e., the spike train power spectrum is independent of temporal frequency and thus "white") because both neuronal variability and tuning were matched to effectively complement natural stimulus statistics (*Carriot et al., 2014*; *Carriot et al., 2017*; *Mitchell et al., 2018*). Our previous study was however limited since we only considered a single class of neurons within the vestibular nuclei and further did not take into account the effects of neural heterogeneities.

Importantly, while VO neurons mediate vestibulo-spinal reflexes as well as self-motion perception as mentioned above, two other distinct neuronal classes instead mediate the vestibulo-ocular reflex (VOR) and its adaptation. Specifically, position-vestibular-pause (PVP) neurons make the primary contribution to the VOR, whereas eye-head (EH) neurons receive cerebellar input and are required for

VOR adaptation and motor learning (see *Cullen, 2012* for review) (*Lisberger, 1984*; *Lisberger, 1994*; *Lisberger et al., 1994*; *Ramachandran and Lisberger, 2008*). The VOR generates robust compensatory eye movements in response to head movements encountered during everyday life in order to stabilize gaze (*Goldberg et al., 2012*) and is an attractive model for understanding how information transmitted by sensory neurons is actually decoded downstream to generate behavior. Importantly, the VOR requires that information as to the detailed timecourse of head movements be contained in the spiking activities of sensory neurons and transmitted to motor areas. However, to date, the effects of neural heterogeneities, particularly in terms of variability, within and across all three vestibular neuronal classes (i.e., PVP, EH, and VO) on coding of naturalistic self-motion remains unknown.

Accordingly, here we investigated how differences in variability within and across distinct central vestibular neuronal classes impact coding strategy during naturalistic self-motion. We first establish that, in the absence of stimulation, all three neural classes displayed a similar wide range of variability as quantified by the coefficient of variation (CV) of the resting discharge. To next determine whether neurons optimally encoded naturalistic self-motion stimuli, we computed their response power spectra as well as the mutual information. Specifically, we tested whether the response power spectrum was independent of frequency or, equivalently, whether the mutual information was close to its maximum possible value for a given level of variability as predicted by theory (*Shannon, 1948*; *Rieke et al., 1996*). We found that, within each class, neurons displaying high resting discharge variability displayed temporal whitening. In contrast, neurons displaying lower resting discharge variability did not display temporal whitening but instead faithfully encoded the stimulus's detailed timecourse as assessed by linear stimulus reconstruction. Interestingly, our results show that faithful encoding was greatest for PVP neurons, suggesting that this coding strategy is necessary to generate the compensatory VOR eye movements observed during naturalistic self-motion stimulation. Using a well-established model of VOR pathways, we validated this prediction. Our findings suggest a novel functional role for variability toward establishing different coding strategies as required for different vestibular functions.

## Results

In order to investigate the effects of differences in variability on coding strategy for naturalistic self-motion by central vestibular neurons, single-unit recordings were made from PVP (N = 20), EH (N = 15), and VO (N = 17) neurons within the vestibular nuclei of macaque monkeys (*Figure 1A*, top). PVP, EH, and VO neurons were identified using standard methodology (reviewed in *Cullen, 2012*) that characterized a given neuron's responses to rotational vestibular stimulation in the yaw axis (1 Hz, 40°/s peak velocity) and eye movements (*Figure 1—figure supplement 1*; see Materials and methods). Specifically, each neuron's sensitivity to sinusoidal vestibular stimulation was recorded in the dark and while monkeys cancelled their VOR by fixating a target that moved with the vestibular turntable (VOR cancellation). Neuronal sensitivity to eye movements was assessed during steady fixation, saccades, and smooth pursuit (see Materials and methods). The vestibular and eye movement sensitivities of PVP, EH, and VO neurons in our dataset are shown in *Figure 1—figure supplement 2* and agreed with previously published values (*Roy and Cullen, 2002*; *Roy and Cullen, 2003*; *Massot et al., 2011*; *Mitchell et al., 2018*). In particular, we note that PVP and EH neurons can be distinguished based on their differential sensitivities to eye and head movements during smooth pursuit and VOR cancellation, respectively (*Figure 1—figure supplement 1*).

During stimulation, there is a component of the variability in the neural response that can be explained by the stimulus and a component that cannot (*Stein et al., 2005*). The latter component contributes to what is known as trial-to-trial variability in the neural response to repeated presentations of a given stimulus. Such trial-to-trial variability is largely determined by the variability of the resting discharge for vestibular neurons (*Sadeghi et al., 2007*; *Massot et al., 2011*; *Jamali et al., 2013*; *Mitchell et al., 2018*). This is because, in order to be detected, head movements must sufficiently perturb the resting discharge. As such, it is much easier to detect stimulation when the resting discharge is more regular (i.e., less variability) than when it is more irregular (i.e., more variability) (reviewed in *Cullen, 2012*).

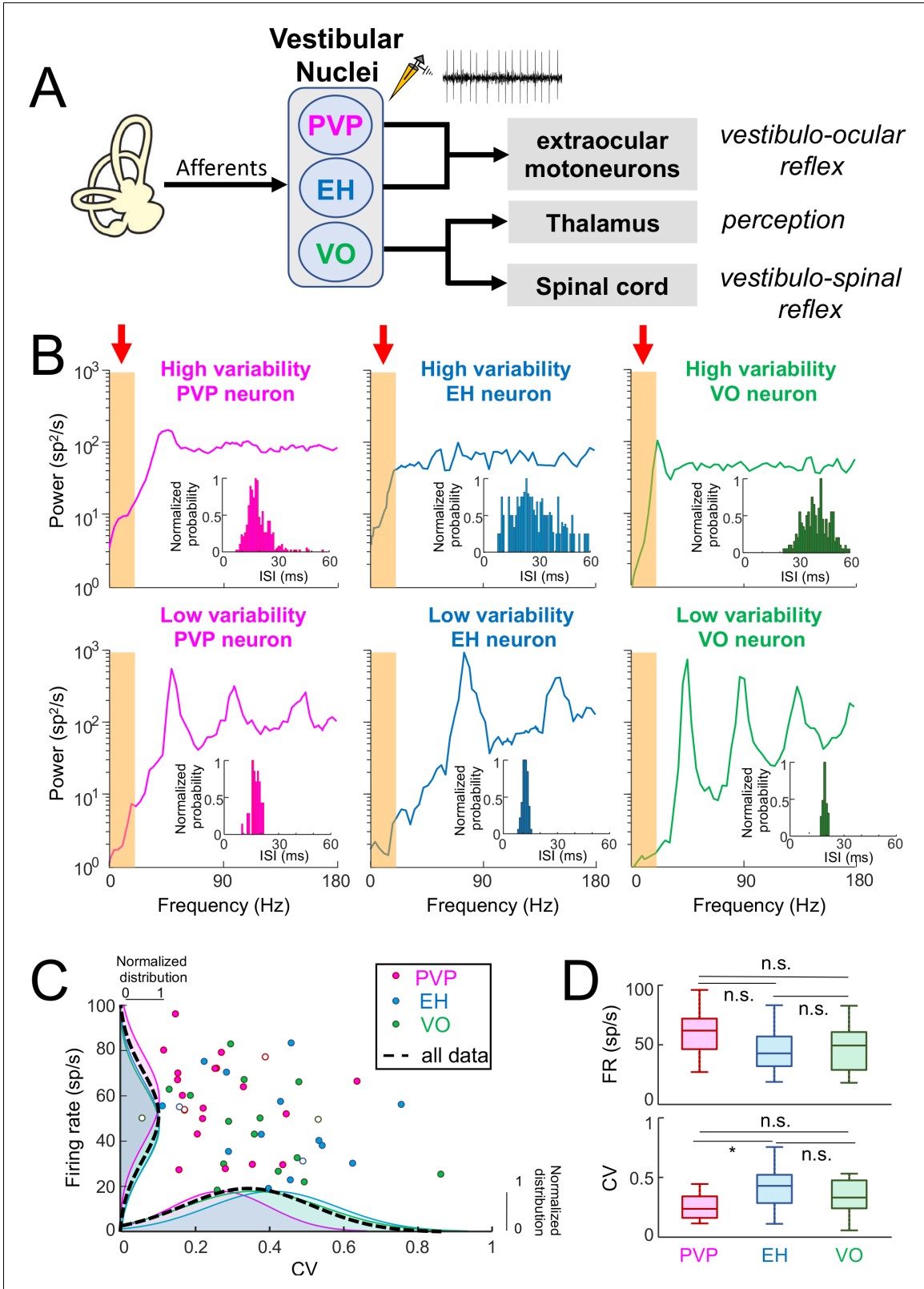

**Figure 1.** Central vestibular neurons display a wide range of variability in the absence of stimulation. (**A**) Afferents from the vestibular periphery project to three classes of neurons found in the vestibular nuclei. Position-vestibular-pause (PVP, magenta) and eye-head (EH, blue) neurons project to extraocular motoneurons within the abducens nucleus and mediate reflexive behaviors such as the vestibulo-ocular reflex (VOR). Vestibular-only (VO, green) neurons project to the ventral posterolateral (VPL) nucleus of the Thalamus, thereby mediating self-motion perception, as well as to the spinal

*Figure 1 continued on next page*

*Figure 1 continued*

cord, mediating vestibulo-spinal reflexes. Recordings were made from PVP, EH and VO neurons. (B) Top: Spike train power spectra of example PVP (left), EH (middle), and VO (right) neurons that display high variability. Bottom: Spike train power spectra of example PVP (left), EH (middle), and VO (right) neurons that display low variability. The orange bands indicated by the red arrows show the frequency range of naturalistic head motion stimuli (0–20 Hz). The insets show the interspike interval (ISI) distribution for each example neuron. (C) Firing rate as a function of the ISI coefficient of variation (CV). In all three cases, there was no significant correlation (PVP: R = −0.08, p=0.71; EH: R = −0.29, p=0.29; VO: R = −0.38, p=0.13). The dashed black curves show the distributions for all data. The six example neurons shown in panel B are represented by open symbols. (D) Top: Population-averaged firing rates for PVP, EH, and VO neurons did not differ significantly from one another (one-way ANOVA, $F_{(2,51)}$ = 2.31, p=0.11). Bottom: Population-averaged CV values for PVP, EH, and VO neurons. The CV of VO neurons was more broadly distributed than that of PVP and EH neurons (Levene's test F = 7.2, p=0.001) while the distribution of the firing rate was similar for all classes of neuron (Levene's test F = 0.87, p=0.43). PVP neurons displayed lower CV values than VO and EH neurons on average (one-way ANOVA, $F_{(2,51)}$ = 3.58, p=0.03).

The online version of this article includes the following figure supplement(s) for figure 1:

**Figure supplement 1.** Neuronal classification.
**Figure supplement 2.** Sensitivities to head and eye movements for PVP, EH, and VO neurons.

## Central vestibular neurons display large heterogeneity in their resting discharges

We quantified the resting discharges of central vestibular neurons using both the mean firing rate as well as the CV of the interspike interval (ISI) distribution (see Materials and methods). The latter measure was used to quantify the resting discharge variability. Our results show that central vestibular neurons displayed resting discharges that were quite variable (PVP: 58 ± 19 sp/s, CV = 0.26 ± 0.13; EH: 47 ± 19 sp/s, CV = 0.40 ± 0.17; VO: 50 ± 18 sp/s, CV = 0.35 ± 0.18). Interestingly, for all three classes, some neurons displayed resting discharges that were more regular as quantified by low CV values, while other neurons instead displayed resting discharges that were more irregular as quantified by higher CV values (*Figure 1B*, insets).

We also quantified the resting discharge of central vestibular neurons by computing the spike train power spectrum (see Materials and methods). There were large differences between the power spectra of neurons displaying low and high resting discharge variability. Indeed, for neurons with low variability, the spike train power spectrum varied strongly as a function of frequency. Notably, spectral power displayed local maxima at the neuron's firing frequency (i.e., the fundamental; *Figure 1B*) as well as higher harmonics (i.e., integer multiples of the fundamental). Spectral power was low within the frequency range of natural self-motion (i.e., 0–20 Hz; see orange bands in bottom panels of *Figure 1B*), which is expected as theory predicts spectral power at low frequencies is proportional to $CV^2$ (*Holden, 1976*). In contrast, the spike train power spectrum of more irregular neurons was more independent of frequency as evidenced from lack of local maxima at higher harmonics (*Figure 1B*, top panels). Spectral power within the frequency range of natural self-motion (see orange bands in bottom panels of *Figure 1B*) was higher than for neurons with low variability. We note that this is again expected based on theory (*Holden, 1976*). It is important to note that the CV distribution was unimodal for all neuron classes (*Figure 1C*, bottom) which suggests that variability is distributed along a continuum for central neurons. We emphasize that we looked at example neurons whose CV was within the lower and higher range of the distribution. For simplicity, we will henceforth refer to these neurons as having "low resting discharge variability" and "high resting discharge variability", respectively.

To assess whether differences in resting discharge variability displayed by central vestibular neurons were related to other discharge properties, we first compared the mean firing rate to CV across our dataset (*Figure 1C*). We found that the mean firing rate was not significantly correlated with CV for all central vestibular neuronal classes (PVP: R = −0.08, p=0.71; EH: R = −0.29, p=0.29; VO: R = −0.38, p=0.13). Further, we did not observe any significant differences between the resting firing rates of PVP, EH, and VO neurons (p≥0.11, one-way ANOVA with Bonferroni correction; *Figure 1D*, top panel) in general, except that PVP neurons displayed significantly lower CV values than EH neurons (p<0.03, one-way ANOVA with Bonferroni correction; *Figure 1D*, bottom panel). Thus, while central vestibular neurons displayed a wide range of resting discharge firing rate and variability within each class, both quantities were similarly distributed for all three classes.

## Effects of variability on central vestibular neuronal coding of naturalistic self-motion

We next investigated the effects of variability on coding strategy during naturalistic self-motion stimulation. To do so, we recorded each neuron's activity during the application of yaw rotations whose time course closely mimicked those experienced during natural conditions (i.e., the naturalistic self-motion stimulus; *Figure 2A,B*, see Materials and methods). The time-dependent firing rates in response to naturalistic self-motion stimulation of two example PVP neurons displaying high (left panel) and low (right panel) variability are shown in *Figure 2C*. Similar graphs for both high and low variability EH and VO example neurons are shown in *Figure 2—figure supplement 1*. These examples were typical in that all neurons responded to the head motion stimulus, although EH neurons typically responded with less modulation in firing rate than PVP and VO neurons (compare *Figure 2—figure supplement 1* to *Figure 2C*). To quantify optimized coding via temporal whitening, we computed the spike train power spectrum during naturalistic stimulation (see Materials and methods). Our analysis revealed that the power spectra of PVP, EH, and VO neurons displaying high variability were independent of frequency (*Figure 2D*, solid curves) as they did not deviate from the Poisson confidence interval with the exception of low (<1 Hz) frequencies for the example EH neuron (*Figure 2D*, gray bands). As mentioned above, the fact that the spectral power was independent of frequency is indicative of temporal whitening. In contrast, spectral power for PVP, EH, and VO neurons displaying low variability decayed similarly to the stimulus power spectrum with increasing frequency and was thus not independent of frequency (*Figure 2D*, compare dashed curves to solid black). As such, these power spectra strongly deviated from the Poisson confidence interval (*Figure 2D*, gray bands), indicating that these neurons did not display temporal whitening.

To quantify differences in temporal whitening, we computed the same whitening index measure that was used in our previous study (*Mitchell et al., 2018*). Overall, we found that the whitening index similarly depended on CV for all neuronal classes (*Figure 2E*, left panel). Further, we found that the whitening index was independent of resting discharge firing rate (*Figure 2—figure supplement 2A*) and the whitening index computed from neurons whose resting discharge firing rates were within a narrow range (45–55 sp/s) strongly depended on CV (*Figure 2—figure supplement 2B*). These results, together with the fact that neural sensitivity was independent of variability (*Figure 2—figure supplement 3*), suggest that the differences in whitening are primarily due to differences in variability and, as such, are universal for central vestibular neurons. To test that changes in whitening index were primarily due to changes in CV, we first built a simple model (see Materials and methods) in which changes in variability were explored systematically. Overall, this model correctly fits data from PVP, EH, and VO neurons across all levels of resting discharge variability (*Figure 2E*, left panel, red curve). Second, to further test our hypothesis, we built a linear-nonlinear cascade model (see Materials and methods and *Figure 2—figure supplement 4A*) that included the effects of variability in order to predict the spike train power spectrum during naturalistic stimulation for individual neurons. Overall, this model accurately predicted the response power spectra of the same six example neurons (*Figure 2—figure supplement 4B*). As such, there was excellent agreement between predicted and actual whitening index values for our dataset (*Figure 2—figure supplement 4C*; Student's t-test, PVP: $F_{(19)}$ = 0.44, p=0.66; EH: $F_{(14)}$ = 0.98, p=0.34; VO = $F_{(16)}$=0.96, p=0.35). We note that the power spectrum of the resting discharge variability was not different from that of the trial-to-trial variability during stimulation for our dataset (*Figure 2—figure supplement 5*). Thus, it is reasonable to model the noise/variability during stimulation by adding the variability from the resting discharge. On average, whitening index values for PVP and EH neurons were comparable to those obtained for VO neurons (*Figure 2E*, right panel). While consistent with our previous finding that, at the population level, VO neurons tend to optimally encode the stimulus via temporal whitening, our current findings emphasize that there are important differences in optimal coding at the individual neuron level both within and across neuronal classes. Specifically, central neurons with low variability tend to not display temporal whitening.

We note that, unlike VO neurons, PVP and EH neurons are sensitive to eye as well as head movements. Thus, we next tested that our results for PVP and EH neurons were not due to patterning of quick phases and/or systematic changes in eye position that occurred throughout vestibular stimulation. To do so, we first compared results obtained on data where spiking activity during vestibular quick phases was removed and the remaining slow phase epochs concatenated (i.e., was is shown in

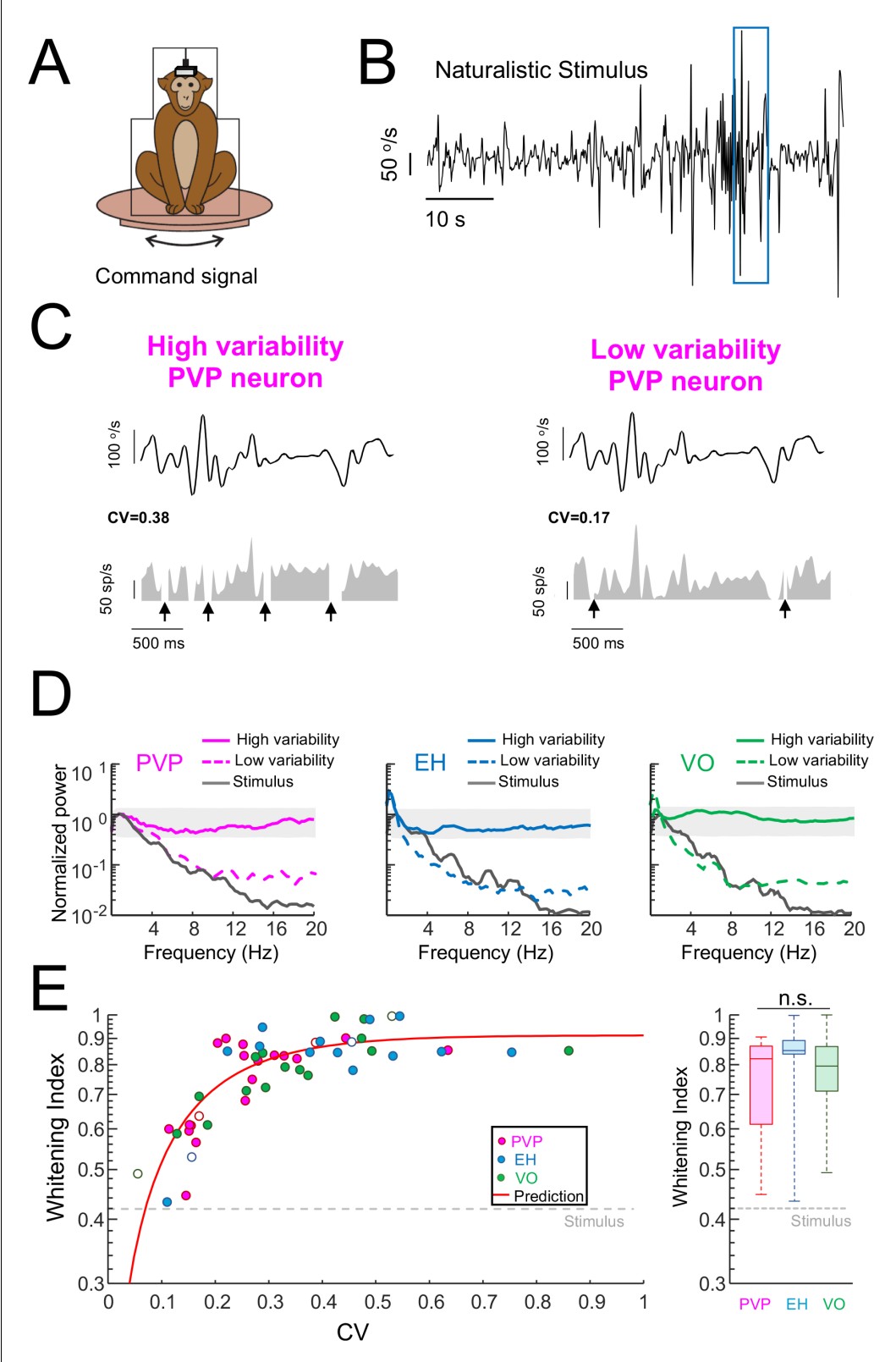

**Figure 2.** Variability strongly influences optimized coding via temporal whitening. (A) The animal was head fixed within a chair that was mounted on top of a turn table whose command signal was adjusted to give rise to head movements that closely matched those seen under natural conditions. (B) Time series showing the entire time course of the naturalistic stimulus. (C) Segment of the naturalistic stimulus (top) corresponding to the blue rectangle in panel B together with time-dependent firing rate responses from the same example PVP neurons shown in *Figure 1* with high (left) and low (right)

*Figure 2 continued on next page*

*Figure 2 continued*

variability. Black arrows indicate times at which vestibular quick phases occurred and during which the neurons paused. (**D**) Response power spectra of PVP (left), EH (middle), and VO (right) neurons with high (solid) and low (dashed) variability. The gray bands show the Poisson confidence interval. It is seen that, while the power spectra of neurons with high variability were always well within the confidence interval and were thus independent of frequency, this was not the case for neurons with low variability whose power spectra decayed as a function of increasing frequency similarly to that of the stimulus (black). (**E**) Left: Whitening index increases as a function of CV. Our model (red line) accurately fits experimental data (all data: $R^2 = 0.63$; PVP: $R^2 = 0.53$; EH: $R^2 = 0.63$; VO: $R^2 = 0.78$). The six example neurons shown in panel B are represented by open symbols. Right: Population-averaged whitening index values were similar for PVP, EH, and VO neurons (Kruskal-Wallis, $H_{(2)} = 2.63$, p=0.26). The gray-dashed lines show the whitening index value computed when using the stimulus' power spectrum.

The online version of this article includes the following figure supplement(s) for figure 2:

**Figure supplement 1.** Example EH and VO neurons with high and low variability.
**Figure supplement 2.** Effects of firing rate on whitening.
**Figure supplement 3.** Gain *vs.* CV for PVP, EH, and VO neurons.
**Figure supplement 4.** Using linear-nonlinear cascade models to predict responses of central vestibular neurons.
**Figure supplement 5.** Resting discharge and trial-to-trial variability for PVP, EH, and VO neurons.
**Figure supplement 6.** Eye position during naturalistic stimuli does not contribute to temporally whitened responses.
**Figure supplement 7.** Signal and noise power, as well as SNR, for example PVP, EH, and VO neurons with high and low variability.

*Figure 2*) to the entire dataset. Overall, no differences were observed (data not shown). Next, we compared results obtained by restricting our analysis to epochs during which the eye position was relatively constant (i.e., within + / - 5°; see Materials and methods) to those obtained using the full dataset (i.e., when eye position was not restricted). No significant differences were observed (*Figure 2—figure supplement 6*; Student's *t*-test, $F_{(34)} = 1.3$, p=0.2). Thus, our analysis indicates that differences in whitening observed for PVP and EH neurons are not due to differences in quick phase generation and/or changes in eye position during stimulation.

Finally, we investigated the relative contributions of variability vs. tuning toward determining temporal whitening for all central vestibular neural classes. Our previous results have shown that variability within the VO neuron population was essential toward determining temporal whitening (*Mitchell et al., 2018*). Here we extended this analysis to all three central vestibular neuron classes and investigated the effects of different levels of resting discharge variability within each class (*Figure 2—figure supplement 7*). Overall, central neurons with lower variability as quantified by the CV of the resting discharge displayed lower noise power than their counterparts with higher variability. However, there were no significant differences in the amount of signal power transmitted. As such, the signal-to-noise ratio (SNR) decreased with increasing variability. This latter result is important as it suggests that central vestibular neurons with lower variability actually transmit more information than those with higher variability. To test this, we computed the mutual information rate (see Materials and methods) for our dataset and found a significant decrease with increasing CV, thereby confirming our hypothesis (*Figure 2—figure supplement 7*). Finally, we quantified optimality of coding (i.e., the mutual information divided by its maximum possible value for a given level of variability) and found a significant increase with increasing CV (*Figure 2—figure supplement 7*), thereby confirming results obtained using the whitening index. Thus, while central vestibular neurons with high variability transmit less information in absolute terms than their counterparts with low variability, they are more optimal in the sense that the mutual information is closer to the maximum possible value.

## Neurons with lower variability faithfully encode the detailed time course of naturalistic self-motion stimuli

Our results so far have shown that central vestibular neurons with low variability did not optimally encode naturalistic self-motion stimuli via temporal whitening but transmitted more information than their counterparts with high variability. This raises the question as to what is the functional role of such neurons? Our results showing that the spike train power spectrum of these neurons decayed with increasing frequency like that of the naturalistic self-motion stimulus, together with higher mutual information rates, suggest that these neurons faithfully relay information about the detailed time course of self-motion signals. Indeed, inspection of their time-dependent firing rate responses to naturalistic self-motion stimulation suggests a strong linear relationship between their responses

and the stimulus (*Figure 2C* and *Figure 2—figure supplement 1*, bottom panels). To test this hypothesis, we quantified the fraction of variance in the stimulus that could be correctly reconstructed by using a linear decoder that minimizes the mean-squared error between the original and reconstructed stimulus waveforms. Specifically, the reconstructed stimulus was obtained by convolving the spiking activity with a filter whose shape was chosen such as to minimize the mean-squared error between the reconstructed and original stimulus waveforms (see Materials and methods and *Figure 3A*). The left panel of *Figure 3B* shows the original naturalistic self-motion stimulus (black) together with the reconstructed stimulus (red) for an example PVP neurons with high variability. The right panel of *Figure 3B* shows the same but for an example PVP neuron with low variability. These examples were typical in that there was a much better match between the original and reconstructed stimuli for the PVP neuron with low variability (*Figure 3B*, compare left and right panels). Qualitatively similar results were obtained for EH and VO neurons (*Figure 3—figure supplement 1*). We quantified these results by computing the coding fraction (CF), which is the fraction of variance in the stimulus that is correctly reconstructed (see Materials and methods). Overall, we found a strong negative correlation between CF and variability as quantified by CV (*Figure 3C*; All data: R = −0.53, p=$10^{-4}$; PVP: R = −0.51; p=0.02; EH: R = −0.07, p=0.8; VO: R = −0.65,p=$4.5\times10^{-3}$). Further, we found that CF was independent of the resting discharge firing rate (*Figure 3—figure supplement 2A*: All data: R = 0.13, p=0.34; PVP: R = 0.05, p=0.82; EH: R = 0.01, p=0.96; VO: R = −0.06, p=0.81). Moreover, CF computed from neurons whose firing rates were within a narrow range (45–55 sp/s) also strongly depended on CV (*Figure 3—figure supplement 2B*: All data: R = −0.54, p=0.03; PVP: R = −0.96, p=0.03; EH: R = −0.20, p=0.7; VO: R = −0.87, p=0.02). These results strongly suggest that changes in CF were primarily due to changes in the resting discharge CV.

Importantly, the fact that low values of CF were obtained for neurons with high variability was not trivially due to a lack of response. This is because, as mentioned above, there was no significant correlation between neural sensitivity and variability (*Figure 3D*). Moreover, as noted above, there was no significant correlation between neural sensitivity to sinusoidal stimulation at different frequencies and variability (*Figure 2—figure supplement 2*). Overall, PVP neurons displayed the largest coding fraction, followed by VO neurons, and EH neurons displayed the lowest coding fractions overall (*Figure 3E*). These results demonstrate that neurons with low variability more faithfully represented the stimulus' detailed time course in their spiking activities than their counterparts with high variability. Thus, together with our results above showing that neurons with low variability do not display temporal whitening, we conclude that variability establishes a trade-off between faithful stimulus encoding and temporal whitening.

Faithful encoding of the time course of naturalistic self-motion stimuli by neurons with low variability is best matched to the known decoding properties of VOR pathways in order to generate the VOR.

Our results show that, across all classes, PVP neurons most faithfully represented the stimulus' detailed time course in their firing activities. Nevertheless, a significant fraction of PVP neurons displayed temporal whitening. Importantly, information is only useful to the organism if it is decoded by downstream brain areas. As mentioned above, VOR pathways are an attractive model for understanding how information transmitted by sensory neurons is actually decoded downstream to generate behavior because of their relatively simple and well-understood neural circuitry. We first tested whether the VOR evoked in response to naturalistic self-motion stimulation was in fact compensatory. We found that indeed both animals generated robust compensatory eye movements characterized by gains approaching unity, such that the eye velocity was essentially the opposite of head velocity (*Figure 4A*). This brings the important question as to how these compensatory eye movements are generated. Specifically, which of faithful encoding or temporal whitening by PVP neurons is most appropriate to generate compensatory VOR eye movements during naturalistic self-motion stimulation.

To answer this question, we used well-established models of VOR pathways in which the head velocity input elicits responses from peripheral vestibular afferents that project to VOR neurons. The output of VOR neurons is in turn decoded by the neural integrator as well as by extraocular motoneurons and the oculomotor plant in order to generate compensatory eye movements (*Figure 4B*, top; see Materials and methods). Our simulations show that, when the naturalistic head velocity stimulus was used as input, the model generated a robust VOR consistent with that observed

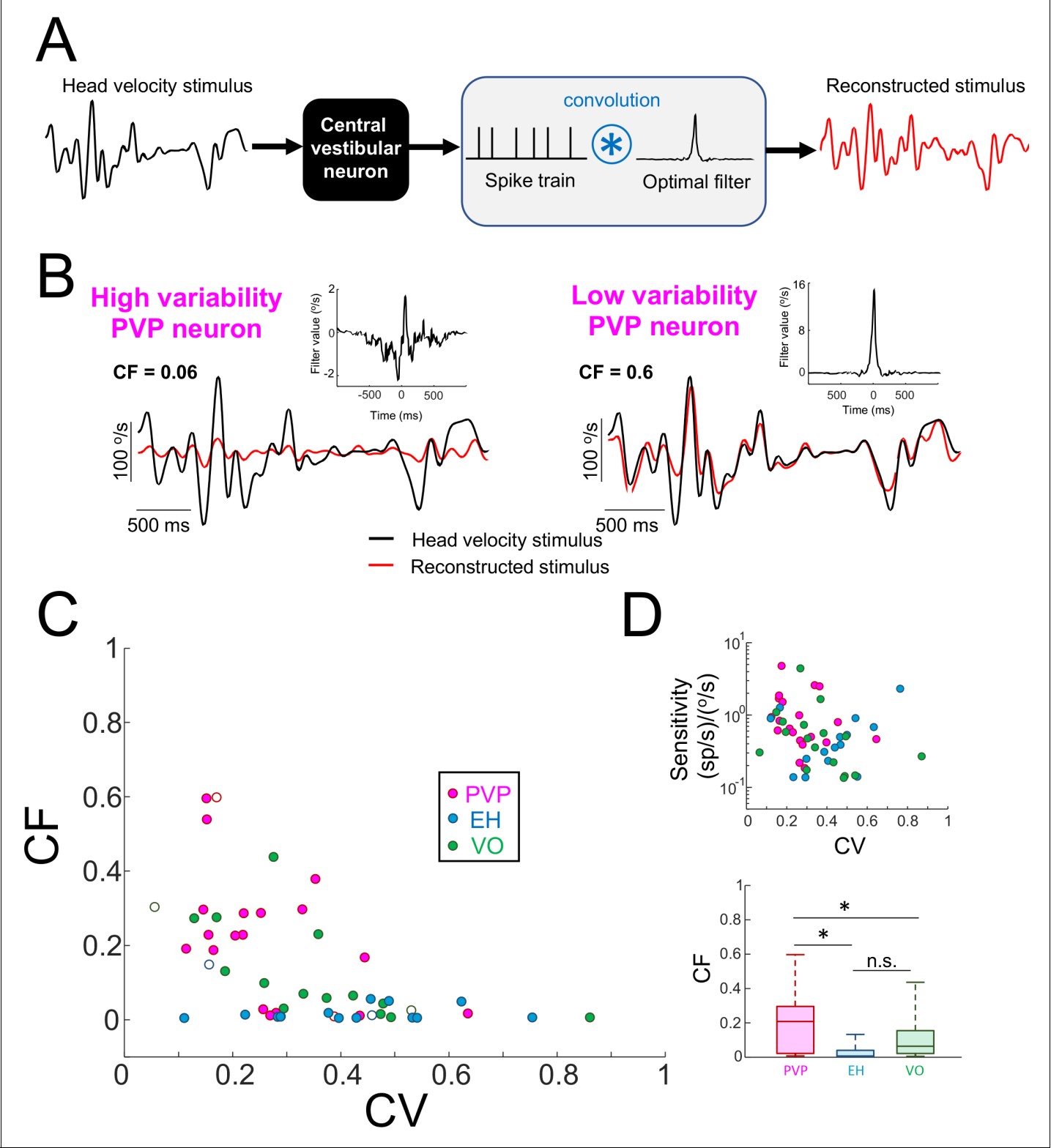

**Figure 3.** Central vestibular neurons with low variability faithfully encode the detailed time course of naturalistic self-motion stimuli. (A) Schematic showing the stimulus reconstruction technique. The head velocity stimulus (left) is presented while neural activity (middle) is recorded and the neuron (middle left) is treated as a "black box". The neural activity is then convolved with an optimal filter (middle right) in order to obtain the reconstructed stimulus (right). This filter is chosen such as to minimize the mean square error between the original and reconstructed stimuli (see Materials and methods). (B) Head velocity (black) and reconstructed (red) stimuli from the same example PVP neurons as in *Figure 1* with high (left) and

*Figure 3 continued on next page*

*Figure 3 continued*

low (right) variability. The quality of the reconstruction was quantified using the coding fraction (CF), which ranges between 0 and 1 and represents the fraction of variance in the stimulus that is correctly reconstructed (see Materials and methods). The insets show the optimal kernel for each example neuron. The gaps in the reconstructed stimulus traces indicate segments during vestibular quick phases. (C) CF decreases as a function of increasing CV (all data: R = −0.53, p=1.0×10$^{−4}$; PVP: R = −0.51; p=0.02; EH: R = −0.07, p=0.8; VO: R = −0.65, p=4.5×10$^{−3}$; R values were computed on log-transformed data). The six example neurons shown in panel B are represented by open symbols. (D) Top: Neural sensitivity did not decrease with increasing CV (all data: R = −0.24, p=0.09; PVP: R = −0.22; p=0.35; EH: R = 0.35, p=0.20; VO: R = −0.25, p=0.34). The legend is the same as in panel C. Bottom: Population-averaged values of CF were highest for PVP and lowest for EH neurons (one-way ANOVA, $F_{(2,51)}$ = 9.1, p=4.5×10$^{−4}$).

The online version of this article includes the following figure supplement(s) for figure 3:

**Figure supplement 1.** Effects of variability on stimulus reconstruction for EH and PVP neurons.
**Figure supplement 2.** Effects of firing rate on coding fraction.

experimentally (*Figure 4—figure supplement 1*). Next, we used the spiking activities of PVP neurons during naturalistic stimulation as input to downstream decoders (i.e., the neural integrator, extraocular motoneurons, and the oculomotor plant) in order to predict compensatory VOR eye movements (*Figure 4B*, bottom). The predicted eye velocity was then compared to the actual eye velocity (i.e., that computed from measured eye movements) for each PVP neuron in our dataset. *Figure 4C* shows the predicted (red) and actual (black) eye velocity time series (inset) and power spectra (main panels) when the spiking activities from example PVP neurons with low (left) and high (right) variability were used as input. We found that there was much better agreement between the predicted and actual eye velocity signals when the input from an example PVP neuron with low variability was used (compare right and left panels in *Figure 4C*). The agreement between predicted and actual eye velocity signals was quantified using a matching index that ranged between 0 (no agreement) and 1 (perfect agreement; see Materials and methods). There was a strong negative correlation between the matching index and variability (R = −0.64, p=2.4×10$^{−3}$; *Figure 4D*, magenta), suggesting that PVP neurons with lower variability that faithfully represent the detailed time course of head movements make the primary contribution to generate compensatory VOR eye movements. To further test this proposal, we compared the performance at stimulus reconstruction from neuron pairs with varying levels of variability. Overall, we found that the performance of neuron pairs with low variability was overall greater than that obtained when considering pairs with low and high variability (*Figure 4—figure supplement 2*), which confirms our proposal. For completeness, we also tested which of faithful encoding or temporal whitening by EH neurons, which are required for VOR adaptation and motor learning, is most appropriate to generate compensatory VOR eye movements. Overall, results for EH neurons were qualitatively similar to those obtained for PVP neurons (*Figure 4—figure supplement 3* and *Figure 4D*, blue; R = −0.64, p=0.01). Thus, taken together, our results point to an important functional role for VOR neurons with low variability that faithfully represent the detailed time course of head movements. Specifically, the activities of these neurons are best matched to the known dynamics of downstream decoders in order to generate compensatory VOR eye movements.

## Discussion

### Summary of results

We investigated the effects of variability on the responses of three different central vestibular neuronal classes: neurons that mediate vestibulo-spinal pathways and project to the vestibular thalamus (i. e., VO), neurons that make the primary contribution to the VOR (i.e., PVP), and neurons that mediate VOR adaptation (i.e., EH). Overall, we found that heterogeneities in resting discharge variability within each class strongly influenced coding strategies. Specifically, neurons with lower resting discharge variability transmitted the highest amounts of information and thus most faithfully encoded the stimulus' detailed timecourse. In contrast, neurons with higher resting discharge variability most optimally encoded the stimulus via temporal whitening, as their mutual information rates were closer to the maximum possible value. These latter neurons displayed lower information rates and thus did not faithfully encode the stimulus. We emphasize that both temporal whitening and faithful encoding were distributed along a continuum for our dataset. Interestingly, we found that PVP neurons, on

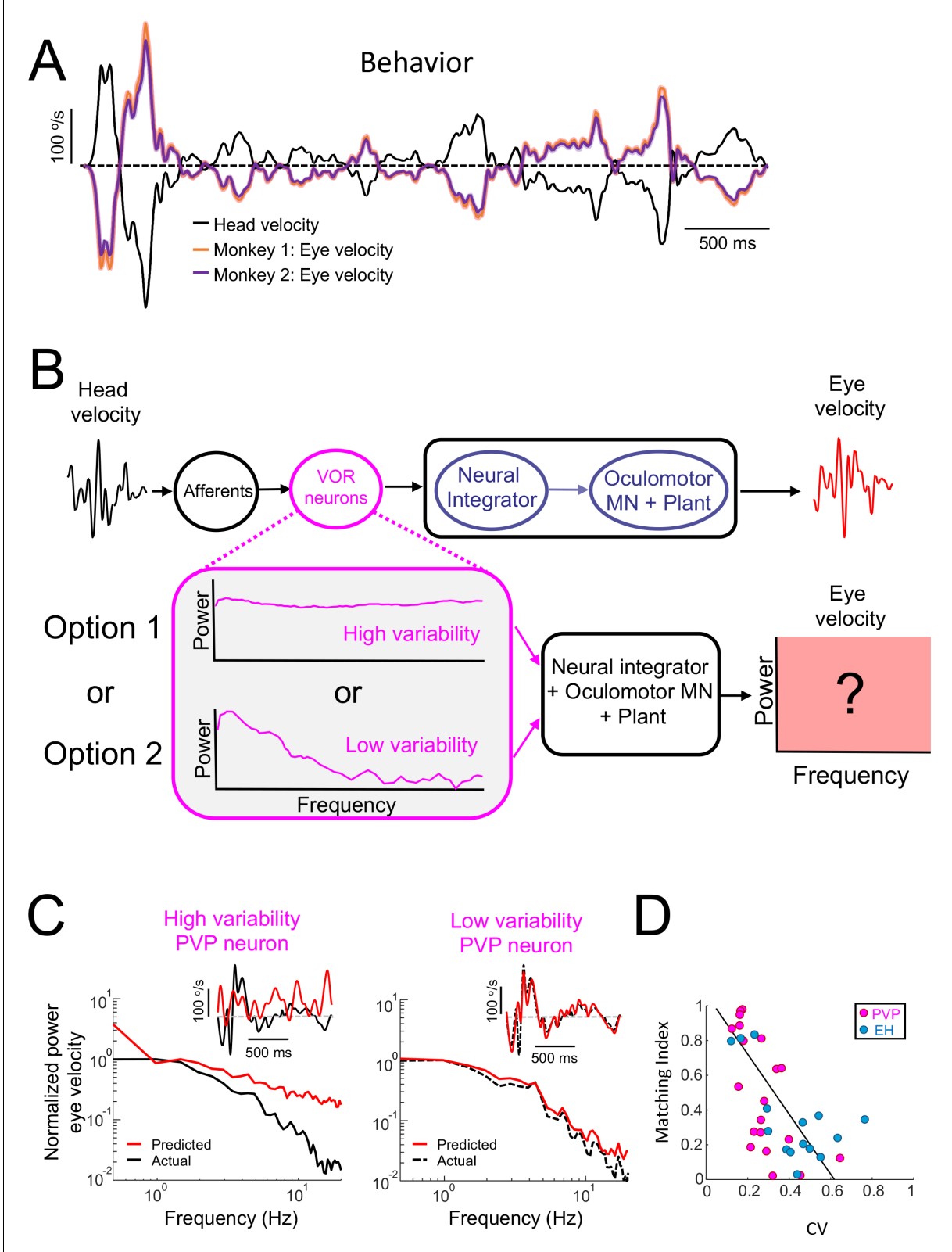

**Figure 4.** Central VOR neurons (i.e., PVP and EH) with low variability are necessary to properly generate compensatory VOR eye movements. (**A**) Head (black) and trial-averaged eye (orange and purple) velocity traces during naturalistic self-motion stimulation for the two animals used in this study. Activity during saccades were not included in the averaging. VOR gain values for both animals were close to unity (VOR gain for monkey 1 = 0.95 ± 0.14; VOR gain for monkey 2 = 0.90 ± 0.12). (**B**) *Top:* schematic showing VOR pathways. VOR neurons receive input from peripheral afferents

*Figure 4 continued on next page*

*Figure 4 continued*

that respond to head movement input and project to the neural integrator as well as extraocular motoneurons ("MN") and the oculomotor plant which generates compensatory eye movements. *Bottom:* We used the spiking activities from both high ("option 1") and low ("option 2") variability VOR neurons as inputs to the neural integrator and oculomotor plant in order to generate predicted eye movements (red box) that were compared with actual eye movement. (C) Predicted (red) and actual (black) power spectra of eye velocity when the input is from a neuron with high (left) and low (right) variability. The insets show the corresponding time series where the dashed gray lines indicate zero velocity. (D) The matching index was negatively correlated with CV (all data: R = −0.65, p=2.9×10$^{-5}$; PVP: R = −0.64, p=2.4×10$^{-3}$; EH: R = −0.64, p=0.01).

The online version of this article includes the following figure supplement(s) for figure 4:

**Figure supplement 1.** Power spectra of the recorded eye velocity (light green) and that predicted from the full VOR model (black) when the recorded head velocity is used as input.

**Figure supplement 2.** Effects of variability on coding accuracy of neuron pairs.

**Figure supplement 3.** Using the power spectra of EH neuron as input in order to generate predictions of the eye velocity.

---

average, more faithfully encoded the stimulus' detailed time course as compared to the two other classes, as quantified by their higher coding fraction values. Using well-established models of VOR pathways, we showed that faithful stimulus encoding, rather than temporal whitening, is required to generate the compensatory VOR eye movements measured in our experiments during naturalistic self-motion stimulation.

## Faithful encoding vs. temporal whitening by VOR neurons, implications for decoding

Here we show for the first time that the detailed time course of head motion can be best recovered from the spiking activities of PVP neurons during naturalistic stimulation. This finding has important implications for understanding the sensory-motor transformations that underlie the generation of the VOR. Indeed, in this context, VOR pathways must account for the dynamics of the oculomotor plant, which are dominated by the visco-elastic properties of the extraocular muscles and passive tissues in the orbit (*Robinson, 1964*; *Enderle and Wolfe, 1988*). Because the properties of the oculomotor plant effectively make it a low-pass filter, the relationship between extraocular motoneuron activity and eye movement must have compensatory frequency dependent dynamics (reviewed in *Robinson, 1981*). More recent studies have shown that the properties of extraocular motoneurons indeed complement those of the oculomotor plant (reviewed in *Cullen, 2012*) during the VOR (*Sylvestre and Cullen, 1999*; *Ramachandran and Lisberger, 2006*). This raises the question as to what the output of PVP neurons should be in order to generate the VOR during naturalistic head movements. Because PVP neurons receive afferent input from the periphery and in turn project directly to extraocular motoneurons to generate compensatory VOR eye movements within 5 ms (*Huterer and Cullen, 2002*), information must be decoded with little or no integration time as achieved by a coding strategy based on faithful stimulus encoding. In contrast, we argue that temporal whitening, which requires filtering of the input and thus more complex decoding strategies with larger integration times, are not suitable for feedforward control of direct VOR pathways.

An open question for understanding the sensory-motor pathways that mediate the VOR is to reconcile how the known nonlinearities displayed by PVP neurons mediate such a linear behavior. Specifically, the VOR shows remarkable linearity in that the resulting eye movements effectively compensate for head movements over a wide range of frequencies and amplitudes (*Huterer and Cullen, 2002*; *Ramachandran and Lisberger, 2006*; *Sadeghi et al., 2006*). In contrast, PVP neurons demonstrate substantial nonlinearities: their spiking activity is silenced for off-direction rotations and demonstrates saturation for on-direction rotations for velocities greater than 200 °/s (*Roy and Cullen, 2004*; *Ramachandran and Lisberger, 2006*). In this context, it is important to note that previous studies have shown that extraocular motoneurons display substantial nonlinearities (*Sylvestre and Cullen, 1999*). It is further conceivable that, for higher amplitude stimuli than the ones considered in the present study, PVP neurons that perform temporal whitening contribute to compensating the nonlinear properties of extraocular motoneurons to ensure a robust VOR.

Interestingly, our results from EH neurons, which mediate VOR adaptation and motor learning (*Lisberger, 1984*; *Lisberger, 1994*; *Lisberger et al., 1994*; *Ramachandran and Lisberger, 2008*), contrast those described above for PVP neurons. Specifically, EH neurons least faithfully followed the naturalistic stimulus' detailed time course as compared to other central vestibular neuronal

classes (*Figure 3D,E*). This result is consistent with previous observations that EH neurons displayed weak sensitivities to artificial vestibular sinusoidal stimuli and are more strongly driven by pursuit signals (*Roy and Cullen, 2003*). Our results show that EH neurons responded to naturalistic head motion such that their spiking activities were temporally whitened. This observation raises the question as to why such an encoding scheme has been adopted in a pathway that was previously shown to be specifically responsible for VOR adaptation and motor learning. We hypothesize that the larger integration time windows needed to properly decode information transmitted via temporal whitening by EH neurons are beneficial for VOR adaptation and motor learning. Further studies are required to test this hypothesis.

## Faithful encoding vs. temporal whitening by VO neurons, implications for decoding

The vestibular system is not only necessary for generating the VOR, but also has multiple other important functions such as self-motion perception and vestibulo-spinal reflexes that control posture. These functions are mediated by a distinct class of neurons within the vestibular nuclei (i.e., VO neurons) that project to the thalamus (*Meng et al., 2007*; *Marlinski and McCrea, 2009*) as well as to the spinal cord (*Abzug et al., 1974*; *Shinoda et al., 1988*; *Gdowski and McCrea, 1999*). Our previous study showed that VO neurons optimally encoded naturalistic self-motion via temporal whitening (*Mitchell et al., 2018*). The present study provides an important addition to this by investigating the effects of variability on coding strategy. Specifically, our new findings reveal that, while the majority of VO neurons displayed high variability and optimally encoded naturalistic head motion stimuli via temporal whitening, a minority of VO neurons displayed low variability and instead faithfully followed the stimulus' detailed time course. This raises the question: why are there differences in coding strategies across the VO neuron population?

We speculate that temporal whitening by VO neurons with high variability is functionally advantageous for the control of vestibulo-spinal reflexes. Notably, the inertia of the head-neck system is higher than that of the oculomotor plant, which has implications for motor control requirements. To date, studies have shown that central vestibular nuclei neurons within vestibulo-spinal pathways are more likely to receive input from afferents with more irregular resting discharges (i.e., irregular afferents), while those within vestibulo-ocular pathways are instead more likely to receive input from afferents with more regular resting discharges (i.e., regular afferents) (*Goldberg et al., 1987*; *Sato and Sasaki, 1993*). Further, the response dynamics of irregular versus regular afferents are best matched to the mechanical demands of the vestibulo-spinal reflex versus the VOR, respectively (*Fernandez and Goldberg, 1971*; *Bilotto et al., 1982*). Recent studies have further shown that irregular but not regular afferents display spike timing precision/phase locking (*Jamali et al., 2016*; *Jamali et al., 2019*). Interestingly, spike timing precision/phase locking was also observed at the next level (*Jamali et al., 2016*). We hypothesize that this single neuron property induces greater synchrony at the VO population level that in turn better compensates for the inertia of the head-neck system (reviewed in *Cullen, 2019*). As such, we propose that temporal whitening by VO neurons with high variability that preferentially receive input from irregular afferents provides enhanced information at the population level, as seen in other systems (*Doi et al., 2012*; *Kastner et al., 2015*). Further studies using multi-unit recordings from VO neurons are needed to test this hypothesis.

While most VO neurons optimally encoded the naturalistic head motion stimulus via temporal whitening, a subset of VO neurons instead faithfully encoded the stimulus. This raises the question: what is the functional role of this subset of neurons? To answer this question, it is useful to consider that VO neurons also project to the thalamus and thus are thought to play a role in self-motion perception and spatial orientation computation (*Meng et al., 2007*; *Marlinski and McCrea, 2009*). One possibility is that the subset of VO neurons that faithfully follows the stimulus' detailed timecourse preferentially projects to thalamus in order to provide information as to the detailed timecourse of head movements. Additional neurophysiological experiments focusing on how neurons within the vestibular thalamus respond to naturalistic head motion stimulation, as well as anatomical studies in which the post-synaptic targets of individual VO neurons are identified, are needed to understand how information transmitted by the VO neuron population is ultimately decoded.

Finally, we note that, throughout this study and our previous publication (*Mitchell et al., 2018*), we assumed that optimality of coding is achieved via temporal whitening. Theoretical studies have shown that temporal whitening gives rise to maximal information if the input SNR is high

(*van Hateren, 1992a*; *Rieke et al., 1996*). In the case where the input noise for a given frequency is high (i.e., a low input SNR), optimal coding is instead achieved by filtering out the neural output at that frequency due to noise contamination (*van Hateren, 1992a*), which has been observed in the retina for weak stimulus intensities (*van Hateren, 1992b*). As such, it is theoretically conceivable that the lack of temporal whitening observed for central vestibular neurons with low variability could be a form of optimized coding that is based on a different constraint than temporal whitening. However, this is unlikely to be the case here since both regular and irregular vestibular afferents that provide input to central neurons display low trial-to-trial variability during naturalistic self-motion stimulation (*Sadeghi et al., 2007*; *Jamali et al., 2016*; *Mitchell et al., 2018*). As such, the input SNRto central vestibular neurons is very likely to be high, such that optimized coding is achieved via temporal whitening according to theory. Moreover, as mentioned above, central neurons within VOR pathways receive input primarily from regular afferents (*Goldberg et al., 1987*; *Sato and Sasaki, 1993*) which display the least trial-to-trial variability during stimulation and thus the highest input SNR (*Sadeghi et al., 2007*; *Jamali et al., 2016*; *Mitchell et al., 2018*). Thus, our results showing that these neurons display the most faithful encoding and the least temporal whitening further support the hypothesis that optimized coding is achieved via temporal whitening for central vestibular neurons.

## Mechanisms underlying differences in variability in central vestibular pathways

Here, we have established that differences in resting discharge variability strongly influence coding strategies by central vestibular neuron populations. This finding then leads to the important question as to what mechanism(s) underlie the different levels of variability observed across all three classes of central vestibular neurons. Notably, our results show that variability is distributed along a continuum for all central neural classes. This is different than for vestibular afferents in which variability displays a bimodal distribution with two distinct classes: regular and irregular (*Goldberg, 2000*). As mentioned above, VOR neurons are more likely to receive input from regular afferents, whereas neurons within vestibulo-spinal pathways are instead more likely to receive input from irregular afferents (*Goldberg et al., 1987*; *Sato and Sasaki, 1993*). Thus, one possibility is that differences in variability across central neurons are due to different amounts of feedforward input from regular and irregular afferents. Another possibility, which is not mutually exclusive, is that differences in the intrinsic properties of central neurons contribute to differences in variability (*Babalian and Vidal, 2000*; *Ris et al., 2001*; *Sekirnjak and du Lac, 2002*; *Kodama et al., 2020*). Finally, it is important to note that central vestibular neurons also receive direct input from central structures including cortical, cerebellar, as well as numerous brain stem nuclei (*Akbarian et al., 1994*; *Voogd et al., 1996*; *McCrea and Horn, 2006*) (see (*Angelaki and Cullen, 2008*) for review). These central inputs also likely contribute to shaping neuronal variability. For example, extracellular recordings in the cerebellar flocculus reveal irregularities in the spontaneous simple spike firing rate of Purkinje cells (*Hoebeek et al., 2005*), which provides a clear source of variability to EH neurons. Further studies are needed to understand how these different sources of input, together with differences in intrinsic properties, contribute to generating the different levels of variability seen experimentally in central vestibular neurons.

## Materials and methods

### Surgical procedures and data acquisition

All experimental protocols were approved by the McGill University Animal Care Committee and complied with the guidelines of the Canadian Council on Animal Care. Two rhesus macaque monkeys (*Macaca mulatta*) were prepared for chronic extracellular recording using aseptic surgical techniques as previously described (*Mitchell et al., 2018*). Briefly, animals were pre-anesthetized with ketamine hydrochloride (15 mg/kg im) and injected with buprenorphine (0.01 mg/kg im) and diazepam (1 mg/kg im) to provide analgesia and muscle relaxation, respectively. Loading doses of dexamethasone (1 mg/kg im) and cefazolin (50 mg/kg iv) were administered to minimize swelling and prevent infection, respectively. Anticholinergic glycopyrrolate (0.005 mg/kg im) was also preoperatively injected to stabilize heart rate and to reduce salivation, and then every 2.5–3 hr during surgery. During surgery, anesthesia was maintained using isoflurane gas (0.8–1.5%), combined with a

minimum 3 l/min (dose adjusted to effect) of 100% oxygen. Heart rate, blood pressure, respiration, and body temperature were monitored throughout the procedure. During the surgical procedure, a titanium post for head immobilization and a titanium recording chambers that allowed access to the vestibular nucleus (VN) were fastened to each animal's skull with titanium screws and dental acrylic. Craniotomy was performed within the recording chamber to allow electrode access to the brain stem. An 18-mm-diameter eye coil (three loops of Teflon-coated stainless-steel wire) was implanted in one eye behind the conjunctiva (*Fuchs and Robinson, 1966*). Following surgery, we continued dexamethasone (0.5 mg/kg im; for 4 days), anafen (2 mg/kg day one, 1 mg/kg on subsequent days), and buprenorphine (0.01 mg/kg im; every 12 hr for 2–5 days). In addition, cefazolin (25 mg/kg) was injected twice daily for 10 days. Animals recovered in 2 weeks before any experimenting began.

During experiments, monkeys were head-restrained and seated in a primate chair mounted on a motion platform rotating about the vertical axis (i.e., yaw rotation). We recorded the single-unit activities of three classes of vestibular neurons within the vestibular nuclei (PVP, EH and VO neurons) using enamel-insulated tungsten microelectrodes. Extracellular activity of the vestibular neurons was initially recorded during standard head-restrained paradigms to characterize their sensitivity to eye movements and head velocity. To quantify the neuronal sensitivity to eye movements, monkeys were trained to visually track a target light. Gaze position was measured using the magnetic search-coil technique. Eye sensitivity was then characterized during saccadic and smooth pursuit eye movements. Both PVP and EH responses were proportional to eye position following saccade and were responsive during smooth pursuit. Characteristic to the PVP, they paused during saccades (*Figure 1—figure supplement 1*). VO neurons were unresponsive to eye movements. Neuronal sensitivities to head velocity were assessed during VOR and vestibulo-ocular reflex cancellation (VORc) paradigms while the monkeys were passively rotated about the vertical axis. During naturalistic yaw rotation, all neuronal classes responded to stimulation in a manner consistent with their classification (*Cullen et al., 1993*; *Cullen and McCrea, 1993*). Sinusoidal head motion stimuli with frequencies 0.5, 1, 2, 3, 4, 5, 8, 17 Hz and amplitudes of 20 deg/s were then applied to characterize head motion sensitivity. We then recorded neural activity during naturalistic head yaw rotation that mimicked the head velocity of a freely moving monkey (*Carriot et al., 2017*), as previously described (*Mitchell et al., 2018*). We note that the probability distribution of the naturalistic head motion stimulus was well-fit by a Gaussian (see *Figure 1—figure supplement 1B* of *Mitchell et al., 2018*). We note that the distribution of head velocities was symmetric around zero (p=0.33, triples test) (*Randles et al., 1980*). Motion platform velocity was measured using a one-dimensional angular gyroscope (Watson Inc). Data were collected through the Cerebus Neural Signal Processor (Blackrock Microsystems). Action potentials were discriminated from extracellular recordings offline by using a custom-written algorithm (Matlab).

## Analysis of neuronal discharges

Data were imported into Matlab for analysis using custom-written algorithms. Head velocity signals were sampled at 1 kHz and digitally low-pass filtered at 125 Hz. For each neuron, we generated a binary spike train $R(t)$ with a sampling rate of 1 kHz. Eye position sensitivities were determined from saccadic as well as smooth pursuit eye movements using standard methodologies (*Roy and Cullen, 2002*). Head velocity sensitivities were then determined during sinusoidal stimulation using standard models (*Roy and Cullen, 1998*). Neuron response spectra during naturalistic stimulation were computed from digitized spike trains using the Matlab function "pwelch" in which epochs during vestibular quick phases were removed. Quick phases were detected using standard methodology (*Roy and Cullen, 2002*). To test whether the neuron's response power spectra was constant across frequencies, we calculated the whitening index as the integral of the spike train power spectrum from 0 to 20 Hz divided by the integral of a simulated white response at maximum neuron power across frequency range (i.e., 0–20 Hz). The stimulus power spectrum and whitening index were computed from the head velocity signal using the same method as for the neuron's response. Periods of spontaneous activity were used to calculate the resting discharge power spectra using the Matlab function "pwelch". For EH and PVP neurons, we concatenated epochs during fixation at 0 deg. Variability was quantified using the CV, which is the standard deviation to mean ratio of the interspike interval (i.e., the times between consecutive action potential firing) distribution.

## Response dynamics for naturalistic stimuli

The response tuning function was computed from the transfer function $H(f)$ using:

$$G(f) = |H(f)|$$

$$H(f) = \frac{P_{RS}(f)}{P_{SS}(f)}$$

Where $P_{RS}(f)$ is the cross-spectrum between the stimulus $S(t)$ and the binary spike train $R(t)$, and $P_{SS}(f)$ is the power spectrum of the stimulus $S(t)$. We used $G(f = 1\ Hz)$ to quantify sensitivity for plotting as a function of variability. Spectral quantities (i.e., power spectra, cross-spectra) were estimated using multitaper estimation techniques (*Jamali et al., 2016*; *Jamali et al., 2019*; *Jarvis and Mitra, 2001*; *Schneider et al., 2015*). We used the stimulus reconstruction method to quantify faithful encoding of the stimulus' detailed time course by neural activity (*Gabbiani and Koch, 1998*; *Marre et al., 2015*; *Massot et al., 2011*; *Rieke et al., 1996*). Specifically, the reconstructed stimulus for N neurons is given by:

$$S_{reconstructed}(t) = \sum_{i=1}^{N} (K_i * R_i)(t)$$

where, for neuron $i$, $R_i(t)$ is the binary sequence and $K_i$ is the optimal kernel. When N=1, the Fourier transform of the kernel $K_1(t)$ is given by the following equation:

$$\tilde{K}_1(f) = \frac{P_{R_1S}(-f)}{P_{R_1R_1}(f)}$$

where, for neuron $i$, $P_{R_iR_i}(f)$ is the power spectrum of the binary spike train $R_i(t)$, and $P_{R_iS}(f)$ is the cross-spectrum between the binary spike train $R_i(t)$ and the stimulus $S(t)$. When N = 2, the Fourier transforms of the kernels are given by:

$$\begin{pmatrix} \tilde{K}_1(f) \\ \tilde{K}_2(f) \end{pmatrix} = \begin{pmatrix} ccP_{R_1R_1}(f) & P_{R_1R_2}(f) \\ P_{R_2R_1}(f) & P_{R_2R_2}(f) \end{pmatrix}^{-1} \begin{pmatrix} P_{R_1S}(-f) \\ P_{R_2S}(-f) \end{pmatrix}$$

where, for neurons $i$ and $j$, $P_{R_iR_j}(f)$ is the cross-spectrum between the binary spike trains $R_i(t)$ and $R_j(t)$. We assessed the quality of the reconstruction by computing the coding fraction CF:

$$CF = 1 - \frac{\sqrt{\varepsilon^2}}{\sigma}$$

where $\varepsilon^2 = \left\langle (S(t) - S_{reconstructed}(t))^2 \right\rangle$ is the mean square error, <...> denotes an average over time, and $\sigma$ is the standard deviation of the stimulus $S(t)$. *CF* ranges between 0 and 1 and represents the fraction of variance in the stimulus that is correctly reconstructed. The stimulus reconstruction was applied to single neurons (i.e., N=1) as well as for PVP neuron pairs (i.e., N=2). For PVP neuron pairs, neurons were grouped into those with "low resting discharge variability" and those with "high resting discharge variability". Specifically, we took the 5 PVP neurons having the lowest resting discharge variability and the 5 PVP neurons with the highest resting discharge variability, as quantified by CV.

We computed the trial-to-trial variability as done previously (*Mitchell et al., 2018*). Briefly, the residuals $\Delta R_i$ were computed as:

$$R_i = R_i - \frac{1}{N}\sum_{i=1}^{N} R_i$$

and the variability power spectrum was computed as the average power spectrum of the residuals. Here $R_i$ is the binary spike train obtained for the $i$th presentation of the stimulus. Our results show that this power spectrum was similar to that of the resting discharge obtained in the absence of stimulation (*Figure 2—figure supplement 3*).

To predict the response power spectrum $P_{RR,predicted}(f)$ to the naturalistic stimulus, we fit a linear-nonlinear cascade model to our data (*Chichilnisky, 2001*), where the predicted firing rate is given by:

$$FR_{predicted}(t) = G\left(\left(\tilde{H} * S\right)(t)\right)$$

where $S(t)$ is the stimulus, $\tilde{H}(t)$ is the Fourier transform of the transfer function $H(f)$, "*" denotes the convolution, and G is a nonlinear function that is determined by plotting the actual firing rate as a function of the linear prediction $\left(\tilde{H} * S\right)(t)$ (*Schneider et al., 2015*). The predicted response power spectrum $P_r(f)$ is then given by:

$$P_{RR,predicted}(f) = P_0(f) + P_{FR,predicted}(f)$$

where $P_{FR,predicted}(f)$ is the power spectrum of $FR_{predicted}(t)$ and $P_0(f)$ is the power spectrum of the binary sequence obtained in the absence of stimulation (i.e., resting discharge). We note that the above mentioned fact that the head velocity stimulus probability distribution is symmetric with respect to zero (i.e., P(s)=P(-s)) implies that there will not be any inconsistencies or biases in our transfer function and LN model estimates (*Meyer et al., 2016*; *Paninski, 2003*; *Chichilnisky, 2001*).

To test whether the response power is independent of frequency $f$, we simulated 1000 Poisson processes with the same number of spikes as contained in each neuron's spike train and computed the power spectra obtained for each Poisson spike train. We found that the distributions at every frequency were Gaussian (Shapiro-Wilk test, all p-values>0.05) as expected from the central limit theorem and obtained a 95% confidence interval that is shown in the figures.

## Contribution of the neuron's eye sensitivity to response power spectrum

PVP and EH neurons are responsive to changes in eye position that occur during naturalistic head motion during the VOR or during quick phases. To test whether the transmitted power is influenced by the neuronal sensitivity to eye position, we computed the response power for concatenated segments of the naturalistic stimulus for which the eye position was confined between $\pm 5°$.

## VOR

VOR gain was computed as the opposite of the slope of the best-fit linear regression between head velocity and eye velocity. Epochs during which the monkeys performed saccades were not included in the regression.

## Model of ocular motoneuron responses to naturalistic self-motion

We used the following model (*Robinson, 2011*) in which the eye velocity is related to the head velocity by the following:

$$\tilde{EV}(f) = T_{afferents}(f)\, T_{VN}(f)\, T_{NI}(f)\, T_{Plant}(f)\, \tilde{HV}(f)$$

Where $\tilde{EV}(f)$ is the Fourier transform of the eye velocity, $\tilde{HV}(f)$ is the Fourier transform of the head velocity, and we have:

$$
\begin{aligned}
T_{afferents}(f) &= \frac{s\,(s\,T_1+1)}{(s\,T_2+1)(s\,T_c+1)} \\
T_{VN}(f) &= -g_{VOR}\frac{T_{VOR}}{(s\,T_{VOR}+1)}\frac{(s\,T_c+1)}{T_c} \\
T_{NI}(f) &= T_{e1} + \frac{1}{s} \\
T_{Plant}(f) &= \frac{s\,e^{-s\tau}}{(s\,T_{e1}+1)(s\,T_{e2}+1)}
\end{aligned}
$$

where $s = 2\pi i f$, $i = \sqrt{-1}$, $T_1$= 0.0175 s, $T_2$ = 0.0027 s, $T_c$=5.7 s are time constants representing the dynamics of sensory transduction and afferent filtering properties (*Hullar et al., 2005*; *Schneider et al., 2015*). $g_{VOR}$ is the VOR gain, $T_{VOR}$=16 s is the VOR time constant. "NI" is the neural integrator. $T_{e1}$= 1 s and $T_{e2}$= 0.016 s are time constants describing the neural integrator and plant dynamics, while $\tau$ = 0.008 s is the delay (*Robinson, 2011*). The output of VOR neurons is given by:

$$\tilde{R}_{VOR}(f) = T_{afferents}(f)\, T_{VN}(f)\, \tilde{HV}(f)$$

where $\tilde{R}_{VOR}(f)$ is the Fourier transform of the spiking activity of VOR neurons. As such, the predicted eye velocity is given by:

$$\tilde{EV}_{predicted}(f) = T_{NI}(f)\, T_{Plant}(f)\, \tilde{R}_{VOR}(f)$$

where $\tilde{EV}_{predicted}(f)$ is the Fourier transform of the predicted eye velocity. Multiplying both sides by the complex conjugate gives us the power spectrum of the predicted eye velocity:

$$P_{EV,predicted}(f) = |\, T_{NI}(f)\, T_{Plant}(f)|^2 P_{VOR}(f)$$

which was then compared with the power spectrum of the eye velocity signal recorded during experiments. The matching index was computed as:

$$MI = \left( 1 - \frac{\left\langle \left( \log\left[ P_{EV,predicted}(f) \right] - \log\left[ P_{EV,actual}(f) \right] \right)^2 \right\rangle}{\sigma_{EP,actual}} \right)$$

where $P_{EP,actual}(f)$ is the power spectrum of the actual eye velocity, $\sigma_{EV,actual}$ is the standard deviation of $\log\left[ P_{EP,actual}(f) \right]$ and "log" denotes the natural logarithm.

## Statistics

Our sample size was comparable to those employed in the field (*Massot et al., 2011*; *Mitchell et al., 2018*). Before statistical analysis, normality of distribution was evaluated using a Shapiro-Wilk's test. Parametric analysis was used (two-tailed t-test or ANOVA) when data were normally distributed. When the data deviated from a normal distribution, non-parametric statistic was performed on the data. All significant effects are reported at $p < 0.05$. The data are available on figshare (http://doi.org/10.6084/m9.figshare.12594803).

## Acknowledgements

This research was supported by the Canadian Institutes of Health Research (JC, KEC, MJC) as well as grants R01-DC002390 and R01-DC018061 from the National Institutes of Health (KEC).

## Additional information

### Funding

| Funder | Grant reference number | Author |
| --- | --- | --- |
| Canadian Institutes of Health Research | 162285 | Jérome Carriot<br>Kathleen E Cullen<br>Maurice J Chacron |
| National Institutes of Health | R01-DC002390 | Kathleen E Cullen |
| National Institutes of Health | R01-DC018061 | Kathleen E Cullen |

The funders had no role in study design, data collection and interpretation, or the decision to submit the work for publication.

### Author contributions

Isabelle Mackrous, Data curation, Formal analysis, Validation, Investigation, Visualization, Methodology, Writing - original draft, Writing - review and editing; Jérome Carriot, Formal analysis, Validation, Investigation, Visualization, Writing - original draft, Writing - review and editing; Kathleen E Cullen, Conceptualization, Supervision, Funding acquisition, Validation, Methodology, Writing - original draft, Writing - review and editing; Maurice J Chacron, Conceptualization, Resources, Software,

Supervision, Funding acquisition, Validation, Methodology, Writing - original draft, Project administration, Writing - review and editing

## Author ORCIDs
Kathleen E Cullen (ID) http://orcid.org/0000-0002-9348-0933
Maurice J Chacron (ID) https://orcid.org/0000-0002-3032-452X

## Ethics
Animal experimentation: All experimental protocols were approved by the McGill University Animal Care Committee (#4096) and complied with the guidelines of the Canadian Council on Animal Care.

## Decision letter and Author response
Decision letter https://doi.org/10.7554/eLife.57484.sa1
Author response https://doi.org/10.7554/eLife.57484.sa2

## Additional files

### Supplementary files
• Transparent reporting form

### Data availability
The data is available on figshare http://doi.org/10.6084/m9.figshare.12594803.

The following dataset was generated:

| Author(s) | Year | Dataset title | Dataset URL | Database and Identifier |
|---|---|---|---|---|
| Mackrous I, Carriot J, Cullen KE, Chacron MJ | 2020 | Data from neural variability determines coding strategies for natural self-motion in macaque monkeys | http://doi.org/10.6084/m9.figshare.12594803 | figshare, 10.6084/m9.figshare.12594803 |

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
