## [Decision Letter]

Thank you for submitting your article "Neural variability determines coding strategies for natural self-motion: implications for perception and behavior" for consideration by *eLife*. Your article has been reviewed by three peer reviewers, one of whom is a member of our Board of Reviewing Editors, and the evaluation has been overseen by Joshua Gold as the Senior Editor. The reviewers have opted to remain anonymous.

The reviewers have discussed the reviews with one another and the Reviewing Editor has drafted this decision to help you prepare a revised submission.

As the editors have judged that your manuscript is of interest, but as described below that substantial additional analyses and clarifications are required before it is published, we would like to draw your attention to changes in our revision policy that we have made in response to COVID-19 (https://elifesciences.org/articles/57162). First, because many researchers have temporarily lost access to the labs, we will give authors as much time as they need to submit revised manuscripts. We are also offering, if you choose, to post the manuscript to bioRxiv (if it is not already there) along with this decision letter and a formal designation that the manuscript is "in revision at *eLife*". Please let us know if you would like to pursue this option. (If your work is more suitable for medRxiv, you will need to post the preprint yourself, as the mechanisms for us to do so are still in development.)

The dichotomy between faithful and efficient coding neurons received a good deal of attention in the consultation among the reviewers. All of the reviewers felt that more needed to be done to clarify and to test this distinction. For example, it is not clear how the decoding errors could be larger for the efficient coding neurons if they really are efficient. Related to this point is whether the efficient coding neurons are simply noisier, and if the whitening comes at the expense of higher noise. Several additional specific points raised in the individual reviews are related to this point. I should emphasize that in consultation the reviewers felt more strongly about this point than is reflected in the individual reviews, and whether you are able to deal with it clearly and effectively will be critical to a re-evaluation of the paper.

Reviewer #1:

This paper follows a previous paper from some of the same authors exploring coding of naturalistic inputs in the vestibular system. The previous paper showed that the combination of stimulus power spectrum, neural filtering and noise led to spectrally-flat ("whitened") responses in one class of vestibular neuron. The present paper extends this analysis to two other types of vestibular neurons, concluding that they show considerable heterogeneity in the degree of whitening. I had several concerns about the analysis/presentation:

Filter calculations and non-Gaussian stimuli:

The paper relies on naturalistic head movement stimuli and uses these for the coding analyses. Some of the calculations presented, however, are valid only for gaussian stimuli. This is particularly true of the linear-nonlinear model and more broadly the construction of the transfer functions. Thus, it seems that the filters extracted could be impacted by the correlation structure of the stimulus. This is a key issue for interpretation of the paper.

Adding noise and optimality:

Neurons that whiten the stimulus are referred to as optimal, while other neurons that encode the stimulus faithfully are referred to as non-optimal. If much of the whitening comes from noise in the resting discharge, however, it does not seem that the "optimal" neurons are really optimal – i.e. they would encode better with non-white responses and less noise. This issue recurs throughout the paper and bears on the tradeoff of optimal vs faithful coding.

How does coding in low and high noise neurons differ:

It would be very useful to see a separation of differences in encoded signal and noise to differences in response power spectra. For example, in Figure 2D, how much of the power spectrum in the high variability neurons is due to noise and how much is due to signal? Related, is the spectrum of the encoded signal similar in high and low variability neurons?

Related to this point, in subsection “Neurons with lower variability faithfully encode the detailed time course of naturalistic self-motion stimuli”: the observation that neurons with low variability encode the stimulus more faithfully than those with high variability is not too surprising. It would help to analyze the nature of the coding – e.g. are there systematic errors or bias in encoding in the case of the high variability neurons? Or are both low and high variability neurons encoding the same temporal frequencies, just with different signal to noise ratios?

Reviewer #2:

This is a very nice extension of previous work, now including more classes of vestibular neurons (PVP/EH/VO), whereas only VO were published on in the earlier work. Some excellent hypotheses are outlined about how different types of signals (faithful stimulus encoders versus efficient whitened outputs) are used in different behaviors. This work nicely connects input statistics to encoding with an eye on behavior.

I have just a few main points that I think would enhance the presentation of these results in the manuscript:

1) A main claim of the paper relies on the dissection of neurons in to high and low CV cells, but it really doesn't seem like the data support a statistical distinction between classes, Figure 1D. Specific questions and comments are:

a) Something else needs to be shown here in Figure 1, like the overall histogram of CV, and the accompanying text should not state that there were significant differences amongst these classes of neurons, then point to a figure in which almost all group comparisons are not significant. This just needs to be made crystal clear throughout: this is a hard-fought-and-won dataset that shows some differences amongst neurons, but there aren't enough data to make claims about broad differences between PVP/EH/VO classes, except for the CF result for PVP vs. EH/VO. The result that the PVP class has a lower CV than EH seems like a very small effect size.

b) This appears to be mostly a paper about high versus low baseline variability (though it would be nice to make rate-matched comparisons, next comment), not about differences in response classes. Put another way, it seems like all classes contain neurons that follow the stimulus and some that perform temporal whitening. The first figure could be reformatted to highlight that more clearly.

c) What is the main claim about subtypes, if the above true? Is it that each type needs a "faithful encoder" channel and a "whitened" channel? It seems that this is exactly what is presented in the Discussion, but then the claims about differences in the EH population and a longer discussion about the VO subtype seems out of place, if that's true. Perhaps PVP can be separated by CF, but the distinction between VO and EH neurons in these data seem more tenuous.

2) Do these results hold if high/low variability neurons are compared in pairs that have similar firing rates?

Reviewer #3:

The authors record from three classes of central vestibular neurons (PVP, EH, VO), which project to different areas. They show that functionally, neurons in each of the classes can be divided into "high variability" (HV) and "low variability" (LV) neurons based on characteristics such as CV of the ISI. HV neurons perform decorrelation (=whitening) consistent with efficient coding, while LV neurons perform "faithful encoding", i.e., permit a precise linear reconstruction of the stimulus. The suggested implication is that this division makes sense in the light of downstream computations: e.g., control of eye velocity in VOR favors LV type encoding, as they show computationally.

I find the paper well-written and well-argued, and would support publication after revisions. I do not see any major technical flaws, but recommend one extra analysis as detailed below.

1) I was confused in how precisely the authors define "High variability" and "Low variability" classes. They look at the FR and CV statistics, and also at the power in the natural frequency band, and all these statistics can be done on natural stim or in the resting state. I presume the classification is based *only* on the CV in the resting state. If that is true, this should be said explicitly (and if you use any thresholds to decide what is "high" and what "low", please specify). Please clarify.

But if the variability is "continuously distributed", then the HV and LV neurons (used as the examples) are only at the extreme ends of that CV distribution. Can you mark where the example neurons of 1B are in Figure 1C (and perhaps in other relevant figures)? For how many of all your neurons do you then see such clear differences between two encoding schemes, what are the neurons in the middle of the CV distribution doing and what is your functional expectation for them? Figures 2E and 3C show the "interpolation" as a function of CV in terms of whitening and CF, showing this continuum of behaviors, but the story seems to tend to much towards a black and white dichotomy between the two extreme behaviors. I would suggest rewording to make clearer that this is not a dichotomy.

2) Figure 2E and 3C show a pretty strong dependence of WI and CF on CV. If the baseline firing rate (that the authors also quantified in Figure 1) is included as an additional explanatory variable for CF in addition to CV, do you see a much better prediction of CF or WI? For example, are low-CV neurons that have a high CF in Figure 3C the ones that have higher firing rates?

3) When the authors interpret the function of HV / LV neurons in Figure 3 and 4, suggesting that LV neurons are better for VOR, they imply that the system is choosing to use one or the other class. But I imagine the population, which is mixed and has a full CV spectrum, is input to the neural integrator. In fact, it would make lots of sense if I look at Figure 3A, to interpret LV neurons as tracking the slower changes in the stimulus well, whereas HV neurons emphasize faster modulation. A system that reads out *both* types of neurons (or the full heterogenous population) is thus expected to perform much better in reconstructing the stimulus and controlling the eyes.

I suggest that the authors test this hypothesis by linearly decoding from two or more neurons. In the easiest scenario, they could take one LV and one HV neuron (maybe examples in Figure 3B) and jointly linearly decode the stimulus. This could be compared with the decoding based on two LV and two HV neurons, to see if the major benefit comes from combination of the two classes. One could reconstruct from even more neurons if that is feasible, to see how the reconstruction depends on the number / type of neurons (cf. Marre et al., 2015), but I don't consider necessary for resubmission. If, however, it turns out that the authors discover a large benefit to decoding really from two classes (LV + HV) jointly, I think the interpretation and discussion needs to be modified: heterogeneity is beneficial since it permits precise full stimulus reconstruction, and so VOR control should not only be done with low variability neurons.

An additional discussion point that the authors may want to consider is that LV neurons can be decoded with more instantaneous filters, whereas decoding from neurons (like HV) that decorrelate requires decoding filters that are extended in time; this may cause delays in the reconstruction and is possibly detrimental if the sensory-motor loop in VOR needs to be fast?

[Editors' note: further revisions were suggested prior to acceptance, as described below.]

Thank you for resubmitting your work entitled "Neural variability determines coding strategies for natural self-motion in macaque monkeys" for further consideration by *eLife*. Your revised article has been evaluated by Joshua Gold (Senior Editor) and Fred Rieke (Reviewing Editor).

The manuscript has been improved but there are some remaining issues that need to be addressed before we can make a final decision, as outlined below:

As you will see below, reviewer #3 has some remaining substantial concerns. In consultation, both reviewers and the Reviewing Editor agreed that these points are quite important and dealing with them fully will be essential if we are to proceed with the paper.

Reviewer #2:

I feel that the paper is much improved after revision. Many of my claims were addressed, as were several of the other reviewers. The Gaussian shape of the naturalistic inputs to this system were clearly a point that needed more emphasis and explanation. I'm glad that's been addressed fully.

The distinction amongst classes of neurons is now much clearer, as is, I believe the Abstract and Results section.

Reviewer #3:

The authors have addressed many of my concerns and their additional analyses have clarified the situation. Especially important are the new quantifications that make it clear that low-CV neurons actually transmit more information (thus better stimulus reconstruction) but do not whiten, whereas high-CV neurons transmit less information due to higher noise power, but their output spectra are white.

I have a two outstanding comments:

1) The authors respond that the self-motion marginal PDF is ~Gaussian with low skew. Even small skew can cause distortions in RF estimates (Meyer et al., 2016); more importantly, the necessary condition for the consistency of RF estimates is spherical symmetry, i.e., P(s) = P(-s) where s is the *full* stimulus waveform, not just a single marginal value (which is what they report). I understand the authors are faced with the empirical issue of a naturalistic stimulus, so I just ask to clarify precisely the conditions for consistent estimation.

2) Perhaps most importantly, although it may appear nitpicking, I would like the authors to go through the text and be very careful about the interchangeable use of "optimal coding" and "whitening", for their high-CV class neurons. I agree that these neurons produce, to a good approximation, a whitened output. I also agree that there is a regime of efficient coding theory, but by no means the *only* operating regime, where the theory predicts as optimal a match between stim statistics, noise, and the neural filter that generates white outputs (van Hateren, 1992a): specifically, this happens at high SNR, where the "input noise" Np (in van Hateren paper) is vanishing (Equation 25). But this is *not* the only regime of optimality. When input noise is high, filters are proportional to (sqrt of) signal spectrum (Equation 30), in this case neurons would not whiten but still be optimal.

In your analysis, you have access to channel noise (Nc in van Hateren notation) which you empirically equate by resting discharge spectrum. But I am not sure if you have direct access to the input noise, Np (I am not familiar enough with the system to know what this would constitute). While the *traditional* regime of application of efficient coding theory is the regime where channel noise dominates over input noise and thus the optimal prediction is whitening, there are cases where the system is not in this regime (retina at low light, or processing of higher order spatial textures beyond V1).

As a consequence, it could be that low-CV neurons are or are not optimal even in the sense of efficient coding, depending on noise constraints we do not know; but they for sure don't whiten.

I would thus recommend being very precise about the claims, e.g., in the Abstract, instead of saying that the neurons did not optimally encode…, I would say they do not whiten. I think it is fair to point out in the paper that the typical regime of efficient coding predicts whitening (van Hateren et al.), but it may be going beyond what you can demonstrate to claim that absence of whitening means non-optimal coding in low-CV neurons. I think that these world-level corrections, to focus on non-whitening vs whitening rather than optimal vs. non-optimal encoding, should not detract from the main message of the paper, and provide an interesting discussion point about optimality.

---

## [Author Response]

The dichotomy between faithful and efficient coding neurons received a good deal of attention in the consultation among the reviewers. All of the reviewers felt that more needed to be done to clarify and to test this distinction. For example, it is not clear how the decoding errors could be larger for the efficient coding neurons if they really are efficient. Related to this point is whether the efficient coding neurons are simply noisier, and if the whitening comes at the expense of higher noise. Several additional specific points raised in the individual reviews are related to this point. I should emphasize that in consultation the reviewers felt more strongly about this point than is reflected in the individual reviews, and whether you are able to deal with it clearly and effectively will be critical to a re-evaluation of the paper.

We understand and appreciate your and the reviewers’ comments regarding optimality vs. faithfulness and agree that this is an important point. To directly address this point in the current manuscript, we have first defined our terms (i.e., optimal vs. faithful coding) in the Introduction. Specifically, optimal coding refers to the fact that the mutual information is at its maximum possible value for a given level of variability or, equivalently, that the output power spectrum is independent of frequency (Shannon, 1946; Rieke et al., 1996). Further, we defined faithful encoding as the spike train containing information as to the detailed timecourse of the stimulus and quantified it by comparing the reconstructed stimulus to the original through the coding fraction (i.e., the relative variance of the stimulus that is correctly reconstructed).

Second, in order to directly address the relationship between faithful encoding and optimal coding in our current dataset, we have performed additional analysis which is shown in Figure 2—figure supplement 7. Specifically, we have separated the response power spectra into their noise and transmitted signal power components and now show these as requested by reviewer 1 in panel A. Further, as requested, we have computed the signal-to-noise ratio and show a strong decrease with increasing variability. As such, the high variability neurons are in fact “noisier” than their low variability counterparts. Further, to clarify what is meant by optimal coding, we computed the mutual information between the stimulus and the spike train and compared it to its maximum possible value for a given level of variability. Our results show that, while the absolute mutual information decreases with increasing CV (Panel B, middle), the relative mutual information (i.e., the ratio of the mutual information to its maximum possible value) actually increases (Panel B, right). Thus, neurons with higher CV, while transmitting less information in absolute terms, are more optimal because the mutual information is closer to its maximum possible value for that level of variability. In contrast, neurons with lower CV transmit more information as to the stimulus’ detailed timecourse (i.e., faithful encoding). These new results are now described and discussed in the revised manuscript. In addition, we address the specific reviewers’ comments below.

Reviewer #1:This paper follows a previous paper from some of the same authors exploring coding of naturalistic inputs in the vestibular system. The previous paper showed that the combination of stimulus power spectrum, neural filtering and noise led to spectrally-flat ("whitened") responses in one class of vestibular neuron. The present paper extends this analysis to two other types of vestibular neurons, concluding that they show considerable heterogeneity in the degree of whitening. I had several concerns about the analysis/presentation:Filter calculations and non-Gaussian stimuli:The paper relies on naturalistic head movement stimuli and uses these for the coding analyses. Some of the calculations presented, however, are valid only for gaussian stimuli. This is particularly true of the linear-nonlinear model and more broadly the construction of the transfer functions. Thus, it seems that the filters extracted could be impacted by the correlation structure of the stimulus. This is a key issue for interpretation of the paper.

We understand the reviewer’s comment and had previously directly addressed this important issue in the original paper (Mitchell et al., 2018 ). Specifically, we have shown that the naturalistic self-motion probability distribution is well-fit by a Gaussian (see Figure 1—figure supplement 1B of Mitchell et al., 2018). We further note that the stimulus probability distribution is symmetric with respect to zero, as evidenced by a low skewness of 0.24. As such, there are no inconsistencies or biases in our transfer function and LN model estimates (Chichilnisky, 2001; Paninski, 2003; see Meyer et al., 2016 for review). This is now mentioned in the Materials and methods.

Adding noise and optimality:Neurons that whiten the stimulus are referred to as optimal, while other neurons that encode the stimulus faithfully are referred to as non-optimal. If much of the whitening comes from noise in the resting discharge, however, it does not seem that the "optimal" neurons are really optimal – i.e. they would encode better with non-white responses and less noise. This issue recurs throughout the paper and bears on the tradeoff of optimal vs. faithful coding.

We understand and appreciate the reviewer’s comment regarding optimality and faithfulness. We agree that this is a major point as mentioned above in our response to the editor’s comments. To directly address this point, we have first clearly defined our terms (i.e., optimal vs. faithful coding) in the revised Introduction. Specifically, optimal coding refers to the fact that the mutual information is close to its maximum possible value for a given level of variability, while faithful encoding refers to the fact the neuron transmits high levels of information about the stimulus, such that the stimulus’ detailed timecourse can be recovered from its spiking activity. Second, we have performed additional analysis of our data to directly demonstrate that neurons with high variability (i.e., HV neurons) are more optimal than neurons with low variability (i.e., LV neurons; see Figure 2—figure supplement 7B, right panel). Indeed, our new results show that the mutual information is closer to its maximum possible value when variability increases. However, the absolute mutual information decreased with increasing variability (Figure 2—figure supplement 7B, middle panel), implying that LV neurons more faithfully encode the stimulus than their HV counterparts. We have revised the text to include description of these new results.

How does coding in low and high noise neurons differ:It would be very useful to see a separation of differences in encoded signal and noise to differences in response power spectra. For example, in Figure 2D, how much of the power spectrum in the high variability neurons is due to noise and how much is due to signal? Related, is the spectrum of the encoded signal similar in high and low variability neurons?

We now show both signal and noise power in Figure 2—figure supplement 7A. The noise power was higher for high variability neurons than for low variability neurons, thereby leading to a signal-to-noise ratio that decreases with increasing variability (Figure 2—figure supplement 7B, left panel). However, both low and high variability neurons displayed similar signal power (Figure 2—figure supplement 7A).

Related to this point, in subsection “Neurons with lower variability faithfully encode the detailed time course of naturalistic self-motion stimuli”: the observation that neurons with low variability encode the stimulus more faithfully than those with high variability is not too surprising. It would help to analyze the nature of the coding – e.g. are there systematic errors or bias in encoding in the case of the high variability neurons? Or are both low and high variability neurons encoding the same temporal frequencies, just with different signal to noise ratios?

As mentioned above, the fact that the stimulus probability distribution is symmetric implies that there are no systematic biases or errors in our estimates. Moreover, as mentioned for the previous comment, our new analysis shows that low and high variability neurons encode the same range of temporal frequencies with similar tuning functions, but that the signal-to-noise ratio is lower for the latter (Figure 2—figure supplement 7B, left panel).

Reviewer #2:This is a very nice extension of previous work, now including more classes of vestibular neurons (PVP/EH/VO), whereas only VO were published on in the earlier work. Some excellent hypotheses are outlined about how different types of signals (faithful stimulus encoders versus efficient whitened outputs) are used in different behaviors. This work nicely connects input statistics to encoding with an eye on behavior.I have just a few main points that I think would enhance the presentation of these results in the manuscript:1) A main claim of the paper relies on the dissection of neurons in to high and low CV cells, but it really doesn't seem like the data support a statistical distinction between classes, Figure 1D. Specific questions and comments are:a) Something else needs to be shown here in Figure 1, like the overall histogram of CV, and the accompanying text should not state that there were significant differences amongst these classes of neurons, then point to a figure in which almost all group comparisons are not significant. This just needs to be made crystal clear throughout: this is a hard-fought-and-won dataset that shows some differences amongst neurons, but there aren't enough data to make claims about broad differences between PVP/EH/VO classes, except for the CF result for PVP vs. EH/VO. The result that the PVP class has a lower CV than EH seems like a very small effect size.

We now plot the firing rate and CV distribution for all three classes pooled in Figure 1 as requested. We have furthermore rewritten the Results to emphasize that the distributions of resting discharge variability and firing rate were similar for all neural classes. Finally, we agree with the reviewer and we now emphasize throughout that variability is distributed along a continuum for our dataset, and that we are showing examples within the lower and higher parts of the range of variability displayed by each class.

b) This appears to be mostly a paper about high versus low baseline variability (though it would be nice to make rate-matched comparisons, next comment), not about differences in response classes. Put another way, it seems like all classes contain neurons that follow the stimulus and some that perform temporal whitening. The first figure could be reformatted to highlight that more clearly.

We agree with the reviewer and we have rewritten this sentence to state that: “Thus, while central vestibular neurons displayed a wide range of resting discharge firing rate and variability within each class, both quantities were similarly distributed for all three classes.”. Thus, while there are strong differences in resting discharge variability within each cell class, there are no major differences across cell classes. Further, we now make rate-matched comparisons as described below in answer to comment 2.

c) What is the main claim about subtypes, if the above true? Is it that each type needs a "faithful encoder" channel and a "whitened" channel? It seems that this is exactly what is presented in the Discussion, but then the claims about differences in the EH population and a longer discussion about the VO subtype seems out of place, if that's true. Perhaps PVP can be separated by CF, but the distinction between VO and EH neurons in these data seem more tenuous.

The reviewer raises an interesting point. Based on our dataset, we show that faithful encoding by PVP neurons is necessary to generate compensatory VOR eye movements as it is best matched to the oculo-motor plant requirements. In contrast, we speculate that the larger integration time window needed to decode temporally whitened information is better suited for VOR adaptation, as seen for EH neurons which displayed poor faithful encoding as compared to the other two classes. For VO neurons, we speculate that temporal whitening is better suited to the mechanical demands of the head-neck system (i.e., head inertia) for vestibulo-spinal reflexes. Further, we propose that faithful encoding by a subset of VO neurons serves to improve perception. To make these points clearer in the revised manuscript, we have rewritten the discussion about PVP/EH neurons. Further, we have shortened the discussion about VO neurons as requested.

2) Do these results hold if high/low variability neurons are compared in pairs that have similar firing rates?

We thank the reviewer for raising this important point. To address it, we have added 2 supplementary figures (Figure 2—figure supplement 2 and Figure 3—figure supplement 2). Our new analysis shows that there is no significant correlation between either of whitening index (Figure 2—figure supplement 2A) or coding fraction (Figure 3—figure supplement 2A) and firing rate. Next, we show that, if we only take neurons whose firing rates are within a restricted range (i.e., 45 to 55 sp/s), there is still a positive correlation between the whitening index and CV (Figure 2—figure supplement 2B), and a significant negative correlation between the coding fraction and CV (Figure 3—figure supplement 2B).

Reviewer #3:The authors record from three classes of central vestibular neurons (PVP, EH, VO), which project to different areas. They show that functionally, neurons in each of the classes can be divided into "high variability" (HV) and "low variability" (LV) neurons based on characteristics such as CV of the ISI. HV neurons perform decorrelation (=whitening) consistent with efficient coding, while LV neurons perform "faithful encoding", i.e., permit a precise linear reconstruction of the stimulus. The suggested implication is that this division makes sense in the light of downstream computations: e.g., control of eye velocity in VOR favors LV type encoding, as they show computationally.I find the paper well-written and well-argued, and would support publication after revisions. I do not see any major technical flaws, but recommend one extra analysis as detailed below.1) I was confused in how precisely the authors define "High variability" and "Low variability" classes. They look at the FR and CV statistics, and also at the power in the natural frequency band, and all these statistics can be done on natural stim or in the resting state. I presume the classification is based only on the CV in the resting state. If that is true, this should be said explicitly (and if you use any thresholds to decide what is "high" and what "low", please specify). Please clarify.

We understand the reviewer’s comment. The quantification of variability is based solely on the CV of the resting discharge activity and this is now clearly mentioned in the results. Further, to avoid confusion, we now refer to “resting discharge variability” throughout the manuscript. We have also emphasized that the distribution of CV is continuous and that we are looking at exemplar neurons whose CV values were within the lower and higher range of this distribution.

But if the variability is "continuously distributed", then the HV and LV neurons (used as the examples) are only at the extreme ends of that CV distribution. Can you mark where the example neurons of 1B are in Figure 1C (and perhaps in other relevant figures)? For how many of all your neurons do you then see such clear differences between two encoding schemes, what are the neurons in the middle of the CV distribution doing and what is your functional expectation for them? Figures 2E and 3C show the "interpolation" as a function of CV in terms of whitening and CF, showing this continuum of behaviors, but the story seems to tend to much towards a black and white dichotomy between the two extreme behaviors. I would suggest rewording to make clearer that this is not a dichotomy.

We understand the reviewer’s comment. First, we now mark the example neurons in Figures 1C, 2E, 3C using open symbols as requested. Further, we have emphasized throughout the results that the variability is distributed continuously and that we are looking at example neurons within the lower and higher range of variability throughout. We also now emphasize that both faithful encoding and optimal coding are distributed continuously in our dataset in the discussion in order to stress that this is not a dichotomy, as requested.

2) Figure 2E and 3C show a pretty strong dependence of WI and CF on CV. If the baseline firing rate (that the authors also quantified in Figure 1) is included as an additional explanatory variable for CF in addition to CV, do you see a much better prediction of CF or WI? For example, are low-CV neurons that have a high CF in Figure 3C the ones that have higher firing rates?

We thank the reviewer for raising this important point. To address it, we have added 2 supplementary figures where we show that there is no significant correlation between either of whitening index (Figure 2—figure supplement 2A) or coding fraction (Figure 3—figure supplement 2A) and baseline firing rate. Next, we show that, if we only take neurons whose baseline firing rates are within a restricted range (i.e., 45 to 55 sp/s), there is still a positive correlation between whitening index and CV (Figure 2—figure supplement 2B), and a significant negative correlation between coding fraction and CV (Figure 3—figure supplement 2B).

3) When the authors interpret the function of HV / LV neurons in Figure 3 and 4, suggesting that LV neurons are better for VOR, they imply that the system is choosing to use one or the other class. But I imagine the population, which is mixed and has a full CV spectrum, is input to the neural integrator. In fact, it would make lots of sense if I look at Figure 3A, to interpret LV neurons as tracking the slower changes in the stimulus well, whereas HV neurons emphasize faster modulation. A system that reads out both types of neurons (or the full heterogenous population) is thus expected to perform much better in reconstructing the stimulus and controlling the eyes.I suggest that the authors test this hypothesis by linearly decoding from two or more neurons. In the easiest scenario, they could take one LV and one HV neuron (maybe examples in Figure 3B) and jointly linearly decode the stimulus. This could be compared with the decoding based on two LV and two HV neurons, to see if the major benefit comes from combination of the two classes. One could reconstruct from even more neurons if that is feasible, to see how the reconstruction depends on the number / type of neurons (cf. Marre et al., 2015), but I don't consider necessary for resubmission. If, however, it turns out that the authors discover a large benefit to decoding really from two classes (LV + HV) jointly, I think the interpretation and discussion needs to be modified: heterogeneity is beneficial since it permits precise full stimulus reconstruction, and so VOR control should not only be done with low variability neurons.

We thank the reviewer for raising this important point. To address, we have split our PVP dataset into neurons with “low variability” and “high variability” as suggested, keeping in mind that variability is actually distributed along a continuum. Specifically, we took the 5 neurons with the lowest values of CV as the “low variability” group and the 5 neurons with the highest values of CV as the “high variability” group. Next, we compared the coding fraction values from all possible pairings between low variability neurons (LVLV), high variability neurons (HVHV), and all possible pairings between low and high variability neurons (HVLV). Overall, the pair-averaged coding fraction for LVHV was significantly lower than that obtained from LVLV pairs (Figure 4—figure supplement 2A). Moreover, the pair-averaged change in coding fraction (i.e., the coding fraction computed from the pair minus the maximum value obtained for individual neurons within that pair) for HVLV was significantly lower than that obtained for LVLV pairs (Figure 4—figure supplement 2B). These results suggest that there is no net advantage in considering mixed neural populations as opposed to populations consisting only of LV neurons for reconstructing the stimulus.

An additional discussion point that the authors may want to consider is that LV neurons can be decoded with more instantaneous filters, whereas decoding from neurons (like HV) that decorrelate requires decoding filters that are extended in time; this may cause delays in the reconstruction and is possibly detrimental if the sensory-motor loop in VOR needs to be fast?

We thank the reviewer for raising this important point and have revised the Discussion accordingly.

[Editors' note: further revisions were suggested prior to acceptance, as described below.]

As you will see below, reviewer #3 has some remaining substantial concerns. In consultation, both reviewers and the Reviewing Editor agreed that these points are quite important and dealing with them fully will be essential if we are to proceed with the paper.Reviewer #3:The authors have addressed many of my concerns and their additional analyses have clarified the situation. Especially important are the new quantifications that make it clear that low-CV neurons actually transmit more information (thus better stimulus reconstruction) but do not whiten, whereas high-CV neurons transmit less information due to higher noise power, but their output spectra are white.I have a two outstanding comments:1) The authors respond that the self-motion marginal PDF is ~Gaussian with low skew. Even small skew can cause distortions in RF estimates (Meyer et al., 2016); more importantly, the necessary condition for the consistency of RF estimates is spherical symmetry, i.e., P(s) = P(-s) where s is the FULL stimulus waveform, not just a single marginal value (which is what they report). I understand the authors are faced with the empirical issue of a naturalistic stimulus, so I just ask to clarify precisely the conditions for consistent estimation.

We understand the reviewer’s comment and agree that the necessary condition for the consistency of RF estimates is spherical symmetry. Our original intent in providing the skewness value was to illustrate that, while non-Gaussian, the distribution of natural head velocities was symmetric with respect to zero. In this revised version, we now explicitly test symmetry by performing a triples test (Randles et al., 1980) and found that we could not reject the null hypothesis (p=0.33). This is now reported in the Materials and methods.

2) Perhaps most importantly, although it may appear nitpicking, I would like the authors to go through the text and be very careful about the interchangeable use of "optimal coding" and "whitening", for their high-CV class neurons. I agree that these neurons produce, to a good approximation, a whitened output. I also agree that there is a regime of efficient coding theory, but by no means the only operating regime, where the theory predicts as optimal a match between stim statistics, noise, and the neural filter that generates white outputs (van Hateren, 1992a): specifically, this happens at high SNR, where the "input noise" Np (in van Hateren paper) is vanishing (Equation 25). But this is not the only regime of optimality. When input noise is high, filters are proportional to (sqrt of) signal spectrum (Equation 30), in this case neurons would not whiten but still be optimal.In your analysis, you have access to channel noise (Nc in van Hateren notation) which you empirically equate by resting discharge spectrum. But I am not sure if you have direct access to the input noise, Np (I am not familiar enough with the system to know what this would constitute). While the traditional regime of application of efficient coding theory is the regime where channel noise dominates over input noise and thus the optimal prediction is whitening, there are cases where the system is not in this regime (retina at low light, or processing of higher order spatial textures beyond V1).As a consequence, it could be that low-CV neurons are or are not optimal even in the sense of efficient coding, depending on noise constraints we do not know; but they for sure don't whiten.I would thus recommend being very precise about the claims, e.g., in the Abstract, instead of saying that the neurons did not optimally encode…, I would say they do not whiten. I think it is fair to point out in the paper that the typical regime of efficient coding predicts whitening (van Hateren et al.), but it may be going beyond what you can demonstrate to claim that absence of whitening means non-optimal coding in low-CV neurons. I think that these world-level corrections, to focus on non-whitening vs whitening rather than optimal vs. non-optimal encoding, should not detract from the main message of the paper, and provide an interesting discussion point about optimality.

We agree that this is an important point. To address it, we have made the requested word changes throughout: specifically, we now make it clear that, when we refer to optimal coding, we mean temporal whitening explicitly. We have furthermore added a new paragraph to the discussion to address the case mentioned by the reviewer for which the input signal-to-noise ratio is low. Specifically, in the vestibular system, input noise to central neurons would come from vestibular afferents. Our previous studies have shown that the trial-to-trial variability displayed by these afferents is low during naturalistic stimulation, as their responses can be predicted solely by considering their tuning (Figure 4—figure supplement 3 of Mitchell et al., 2018), which implies that the input signal-to-noise ratio is high. As such, it is unlikely that the deviation from whitening observed is a signature of optimized coding based on a low input signal-to-noise ratio as seen elsewhere. This argument is made in the Discussion.